# Single-cell analysis of Barrett's esophagus and carcinoma reveals cell types conferring risk via genetic predisposition

## Graphical abstract

## Authors

Marten C. Wenzel, Pouria Dasmeh, Patrick S. Plum, ..., Seung-Hun Chon, Johannes Schumacher, Axel M. Hillmer

## Correspondence

johannes.schumacher@uni-marburg.de (J.S.),
ahillmer@uni-koeln.de (A.M.H.)

## In brief

Esophageal adenocarcinoma (EAC) often progresses from Barrett's esophagus (BE), a complication of gastroesophageal reflux disease. Wenzel et al. studied cells from EAC and BE tissues and found a set of cell types that show strong enrichment of active genes at genetic risk variants and thus contribute to EAC development.

## Highlights

- EAC is driven to a greater extent by local cellular processes than BE development

- Intestinal metaplasia cells can contribute through genetic risk variants to EAC

- These cells have goblet cell-like features and express intestinal stem cell gene *OLFM4*

Wenzel et al., 2025, Cell Genomics 5, 100980
October 8, 2025 © 2025 The Author(s). Published by Elsevier Inc.

# Cell Genomics

CellPress

## Article

# Single-cell analysis of Barrett's esophagus and carcinoma reveals cell types conferring risk via genetic predisposition

Marten C. Wenzel,[1,13] Pouria Dasmeh,[2,6,7,13] Patrick S. Plum,[1,3,4,13] Ann-Sophie Giel,[2,13] Sascha Hoppe,[1] Marek Franitza,[5] Christoph Jonas,[1] René Thieme,[4] Yue Zhao,[3] Dominik Heider,[8,9] Claire Palles,[10] Rebecca Claire Fitzgerald,[11] Christiane J. Bruns,[3] Reinhard Buettner,[1] Alexander Quaas,[1] Ines Gockel,[4] Carlo Maj,[2] Seung-Hun Chon,[3] Johannes Schumacher,[2,*] and Axel M. Hillmer[1,12,14,*]

[1]Institute of Pathology, Faculty of Medicine and University Hospital Cologne, University of Cologne, Cologne, Germany
[2]Institute of Human Genetics, University Hospital of Marburg, Marburg, Germany
[3]Department of General, Visceral, Cancer and Transplantation Surgery, Faculty of Medicine and University Hospital Cologne, University of Cologne, Cologne, Germany
[4]Department of Visceral, Transplant, Thoracic and Vascular Surgery, University Hospital of Leipzig, Leipzig, Germany
[5]Cologne Center for Genomics, University of Cologne, Cologne, Germany
[6]Department of Chemistry and Chemical Biology, Harvard University, Boston, MA, USA
[7]Institute for Evolutionary Biology and Environmental Studies, University of Zurich, Zurich, Switzerland
[8]Department of Data Science in Biomedicine, Faculty of Mathematics and Computer Science, Philipps-University of Marburg, Marburg, Germany
[9]Institute of Medical Informatics, University of Münster, Münster, Germany
[10]Institute of Cancer and Genomic Sciences, University of Birmingham, Birmingham, UK
[11]Medical Research Council (MRC) Cancer Unit, Hutchison-MRC Research Centre, University of Cambridge, Cambridge, Cambridgeshire, UK
[12]Center for Molecular Medicine Cologne, University of Cologne, Cologne, Germany
[13]These authors contributed equally
[14]Lead contact
*Correspondence: johannes.schumacher@uni-marburg.de (J.S.), ahillmer@uni-koeln.de (A.M.H.)

## SUMMARY

Inherited genetic variants contribute to Barrett's esophagus (BE) and esophageal adenocarcinoma (EAC), but it is unknown which cell types are involved in this process. We performed single-cell RNA sequencing of BE, EAC, and paired normal tissues and integrated genome-wide association data to determine cell-type-specific genetic risk and cellular processes that contribute to BE and EAC. The analysis reveals that EAC development is driven to a greater extent by local cellular processes than BE development and suggests that one cell type of BE origin (intestinal metaplasia cells) and cellular processes that control the differentiation of columnar cells are of particular relevance for EAC development. Specific subtypes of fibroblasts and endothelial cells likely contribute to BE and EAC development, while dendritic cells and CD4+ memory T cells seem to contribute to BE development. The diagnostic value of markers characterizing the cell types and cellular processes should be explored for EAC prediction.

## INTRODUCTION

Esophageal adenocarcinoma (EAC) is often preceded by Barrett's esophagus (BE), a benign metaplasia of the epithelium, increasing the risk of developing EAC by 10- to 50-fold.[1,2] BE is associated with gastroesophageal reflux disease (GERD) and is likely connected to a chronically inflamed environment.[3]

BE can undergo histological changes ranging from low- to high-grade dysplasia. At an approximate rate of 6 per 100 patient years, patients progress during the first years of follow-up from high-grade dysplasia to EAC, which has a devastating 5-year survival rate.[4,5] The incidence of EAC increased dramatically

over the last decades,[6] emphasizing the need to better understand the cellular processes that determine which patients with BE progress to EAC.

Most single-cell RNA sequencing (scRNA-seq) studies in the context of BE and EAC have focused on the in-depth description of BE in order to investigate the cell of origin of BE. While the cell of origin of BE has been investigated by Owen et al.[7] and Nowicki-Osuch et al.,[8] our understanding of which other cell types beyond the metaplastic cells contribute to EAC progression is limited.

In the present study, we performed scRNA-seq of the normal esophagus and gastric fundus of nine patients. Five of them were

**Figure 1. Single-cell RNA sequencing of biopsies of the esophagus and gastric fundus**

(A) Graphical description of the experimental setup. Samples were taken from healthy esophageal and stomach tissue of every patient with EAC and every patient with BE, respectively. Depending on the diagnosis, samples of either EAC or BE were taken as a third sample of each patient. All three tissue types per patient were dissociated separately. Single cells were prepared for fluorescence-activated cell sorting (FACS) and RNA sequencing using the 10× Genomics Chromium technology, with one library per sample (three per patient). The data were analyzed using the Seurat platform, among others. The light red esophagus region close to the stomach represents BE, and the dark red region represents EAC. Blue circles indicate sampling strategy for patients with EAC and green circles for patients with BE.

(B) Uniform manifold approximation and projection (UMAP) representation of the whole dataset. Each data point represents a single cell with coloring according to initial cell-type family annotation.

patients with BE, from whom the two normal tissue types and the BE tissues were analyzed, and four patients had EAC, from whom EAC tissues were analyzed in addition to the normal tissues. We integrated our scRNA-seq data and data of a genome-wide association study (GWAS) of BE/EAC[9] to determine which cell types show enrichment for the expression of genes that are located at BE and/or EAC risk loci. We identify specific cell types and cellular processes that contribute to BE and to EAC progression based on germline genetic risk.

## RESULTS

### scRNA-seq of BE, EAC, and normal tissues

We performed scRNA-seq to characterize different cell types of BE and EAC tissues and compared them with the normal esophagus and gastric fundus of the same patients (Table S1). We collected tissue samples in a standardized manner regarding the distances of sample sites and then dissociated them into single cells and sequenced for their RNA content using gel bead-in-emulsion (GEM) technology (10× Genomics; Figure 1A).

After quality control and clustering, we used the expression of typical cell-type marker genes to annotate identified cell types (Figures 1B and S1–S4). The group of epithelial cells could be identified by the expression of the marker gene *EPCAM*. The associated clusters contained cells from BE and EAC but also from the normal esophagus and fundus (Figures S2–S4). We focus in the following on the epithelial cells since BE and EAC are epithelial-derived conditions. For the characterization of

mesenchymal and immune cells, see the STAR Methods and Figures S5 and S6.

To get a better understanding of the features of the epithelial cells, we used the clusters identified as epithelial cells to run a separate analysis (Figures 2A–2D). After initial unsupervised clustering, differences between the analyzed clusters were examined, which justified merging single unsupervised clusters to functionally more coherent groups (Figures S7 and S8; Table S2), allowing the identification of generic cell types based on the top differentially expressed genes.

The clusters later annotated as cancer cells were found to be largely homogenous regarding patient origin, in contrast to the other clusters (Figure 2B). In addition, the epithelial cells of BE and fundus origin showed similarities across patients (foveolar cells-1 and enteroendocrine cells), as did the epithelial cells of BE and EAC origin (intestinal metaplasia cells; Figure 2C). While most non-cancer cell types were derived from both patients with BE and patients with EAC, cancer-associated fibroblasts (CAFs) were exclusively found in EAC patient tissue, and BE fibroblasts and esophageal fibroblasts-2 were exclusively found in BE patient tissue. Basophils and pre-B cells were more abundant in EAC patient tissue (Table S2; STAR Methods).

To quantify proliferative activity across epithelial clusters, we assigned each cell to a cell cycle phase. We followed the strategy presented by Tirosh et al.[10]; determined a cell cycle phase, i.e., G1, S, or G2/M (Figure S8C); and calculated a relative proliferation index by dividing the number of cells in S and G2/M phases by the number of cells in G1 within a given cluster. EAC-02 and EAC-04

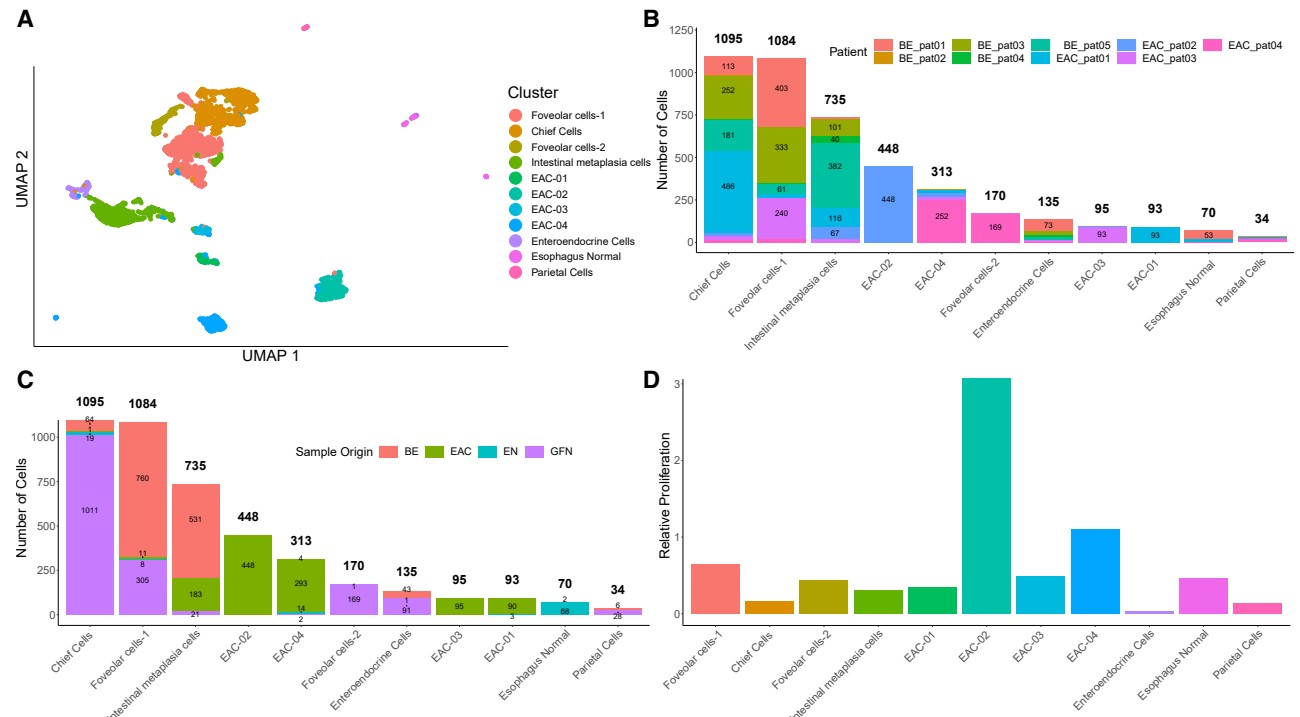

**Figure 2. Epithelial cells show similarities between BE and gastric fundus cells and dissimilarities between EAC cells of different patients**

(A) UMAP representation of the epithelial clusters. Notably, the tumor clusters lie apart from each other with a high degree of homogeneity regarding patient origin.

(B) Composition of epithelial clusters regarding patient origin.

(C) Composition of epithelial clusters regarding sample origin.

(D) Clusters EAC-02 and EAC-04 show the highest relative proliferativeness.

had the highest relative proliferation rate among all epithelial clusters, followed by foveolar cells-1 (Figure 2D).

## Characterization of epithelial cells and detection of typical genomic aberrations in EAC

Next, we annotated the epithelial clusters as chief cells, parietal cells, esophagus normal cells, and enteroendocrine cells based on the expression of marker genes (Figure 3A; STAR Methods; Tables S3 and S4). We identified a group of cells that showed signs of goblet cell differentiation, which defines intestinal metaplasia,[11] but also expression of the gastrointestinal stem cell marker *olfactomedin 4* (*OLFM4*),[12,13] setting this cell type close to the undifferentiated BE cell type implicated by Nowicki-Osuch et al.[8] to be the cell of origin of EAC. In the following, we describe this cell type as intestinal metaplasia, which consisted of cells from both BE and EAC tissue samples. Intestinal metaplasia cells showed a marked expression of *HNF4A*, which likely led to the increased expression of *TFF3* and *LEFTY1*, but not *MYC*, that Nowicki-Osuch et al. observed for undifferentiated BE cells (Figures 3A and S9B). *TFF1*, *TFF2*, and *MUC5AC* were highly expressed in foveolar cells-1 and -2, intestinal metaplasia cells, and cancer cell cluster EAC-01. In addition, *MSMB* was highly expressed in clusters foveolar cells-1 and -2 but was almost absent in cancer cell-containing clusters and intestinal metaplasia cells. EAC-02 and weaker EAC-04 almost exclusively expressed *WNT11*. *CEACAM6* was present in all cancer clusters

and also in intestinal metaplasia cells, with the highest expression in EAC-01.

To discern tumor cells from benign cells, we estimated cluster-specific somatic copy-number alterations (SCNAs) based on single-cell transcriptomic data as described by Tirosh et al.[10] using the R package inferCNV. We found that tumor cells bore distinct patient-specific SCNA profiles in contrast to benign cells (Figure 3B). EAC-01, EAC-02, EAC-03, and EAC-04 showed typical EAC SCNAs[14,15] (Table S5). Notably, EAC-02 and EAC-04 shared three copy gains covering large regions on 6p, 8q, and 9q. In contrast, the group of foveolar and intestinal metaplasia cells did not show characteristic cancer alterations.

## Functional analysis of epithelial cells reveals differences between cells from BE and across EAC tumors

To get a better understanding of the functional properties of the epithelial cell types and subgroups, we used selected cell-type-specific gene sets from Busslinger et al.[16] (Figure 4A, top heatmap) as well as other gene sets regarding benign and malignant esophageal cells or samples (Figure 4A, bottom heatmap) for gene set enrichment analysis (GSEA). Unsupervised clustering of the results showed a qualitative contrast between cells from EAC-01/EAC-03 and EAC-02/EAC-04. EAC-01/EAC-03 demonstrated an enrichment of terms related to gastric pit cells, esophageal late suprabasal cells, and esophageal *KRT6B+*

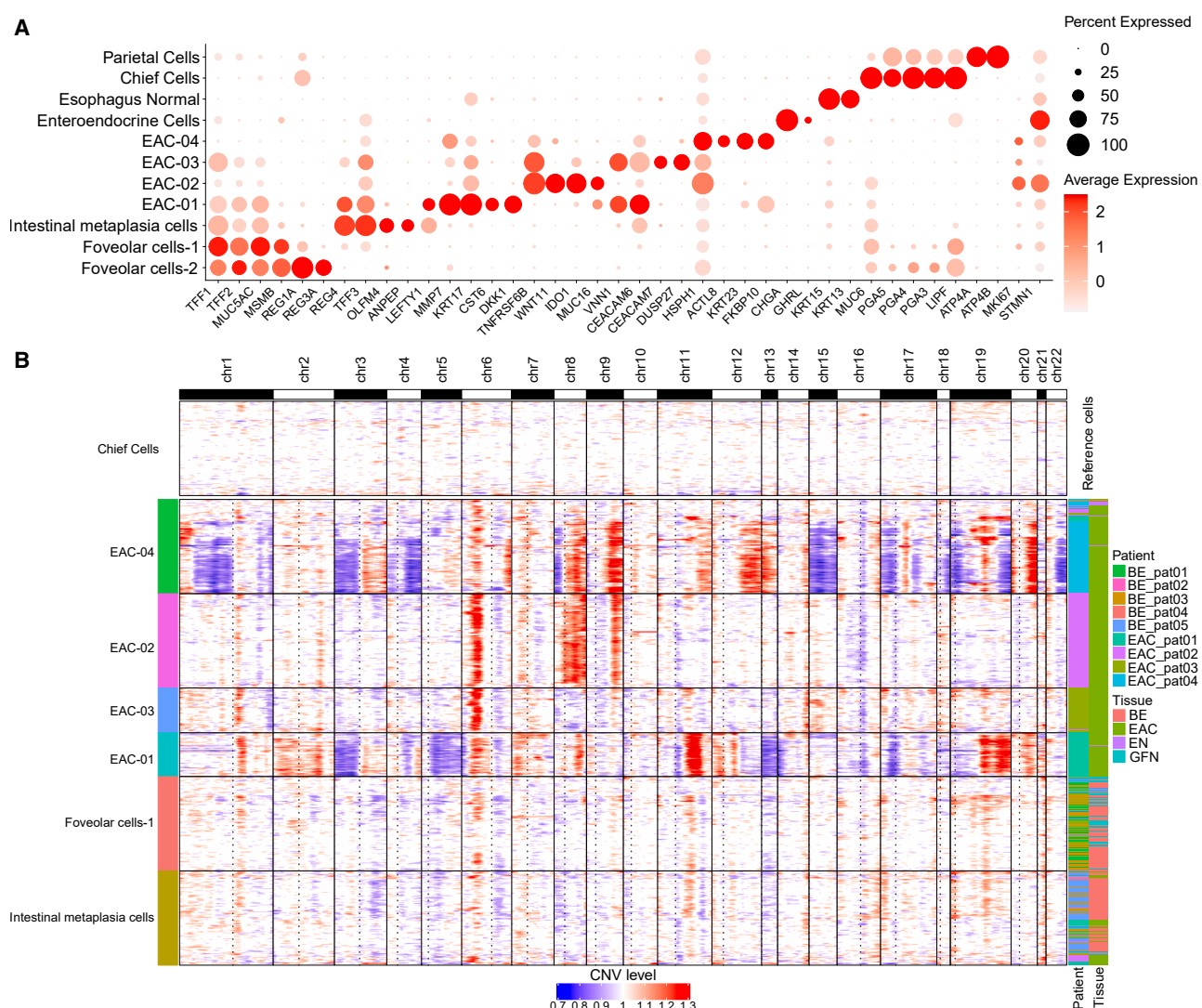

**Figure 3. Characterization of epithelial cell types and identification of EAC-typical SCNA profiles and relative similarities**
(A) Top marker expression (x axis) per epithelial cluster (y axis).
(B) On the x axis, genes are ordered regarding their physical location in the genome. Full lines indicate the chromosomes and dashed lines the approximate position of the centromere. On the y axis, single cells from selected epithelial clusters are shown. The SCNA level is calculated relative to chief cells, which were used as a reference.

secretory progenitor cells from Gao et al.[17] (Figure 4A). In contrast, EAC-02/EAC-04, which were characterized as highly proliferative cells (Figure 2D), exhibited an exclusive enrichment of esophageal early suprabasal cells (Figure 4A).

The three cell types, foveolar cells-1 and -2 and intestinal metaplasia cells, showed a similar enrichment for the gene sets of gastric pit cells and other cell-type signatures from the stomach (Figure 4A), emphasizing the functional similarity of cells from BE and GFN. Notably, only foveolar cell-1 was enriched for the gene set *WANG BARRETTS ESOPHAGUS AND CANCER UP* from Wang et al.[18] (Figure 4A). In the latter study, these genes were upregulated in BE and EAC compared to normal esophageal tissue, giving evidence of an expression-based similarity between this cell type and disease progression.

In contrast, intestinal metaplasia cells do not share these BE-characteristic features and show fewer features of chief and pre-zymogenic cells (Figure S9; Table S6).

Using a different approach, we analyzed the results of a GSEA regarding Gene Ontology (GO) gene sets. For this, the other cell types of the initial analysis were added in order to characterize epithelial cells relative to other cell types (Figure S10). Regarding epithelial cancer cells, again, a segregation of the tumor cells into two groups was observed. EAC-01/EAC-03 were enriched in GO terms related to cell-matrix and cell-cell adhesion, cell migration, and locomotion, while EAC-02/EAC-04 were enriched in terms related to translational processes. In accordance with the findings above (Figure 2D), EAC-02 also showed enrichment in G2-to-M-phase transition.

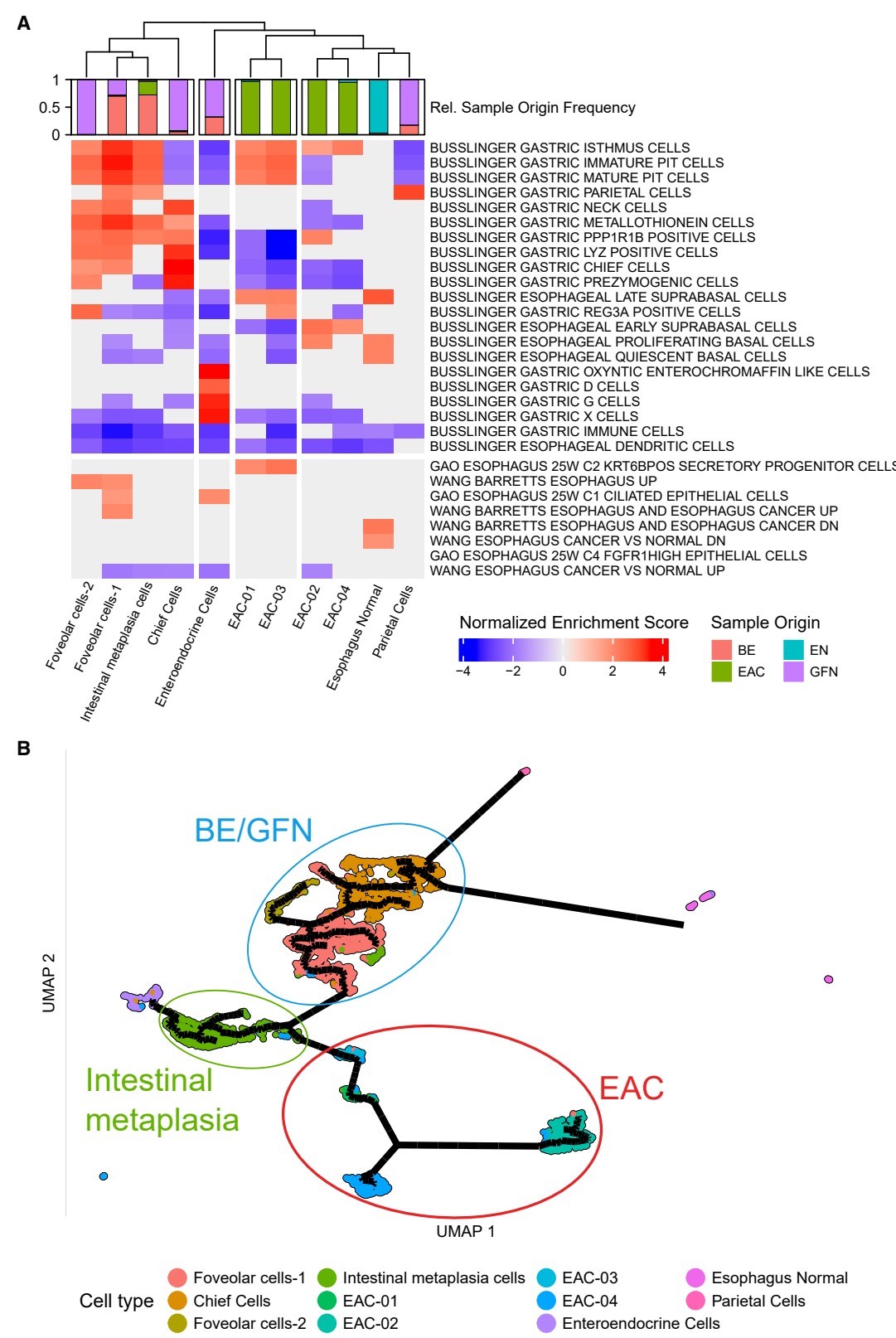

*(legend on next page)*

## Trajectory analysis implies a route to BE and EAC

To investigate the developmental relationship among epithelial cell types, we performed a trajectory analysis using monocle3 based on transcriptomic similarity. Our analysis placed gastric chief cells close to foveolar cells-1 originating from gastric fundus and BE tissue, leading to the group of intestinal metaplasia cells, which consists of cells with goblet cell differentiation markers, but also expressing the intestinal stem cell marker *OLFM4*. The trajectory closely connected foveolar cells-1 with intestinal metaplasia cells, suggesting foveolar cells-1 as the cell of origin for BE (Figure 4B). This is in agreement with the columnar non-squamous phenotype of BE cells and supports the model of gastric cardia cells, located next to the fundus, as the origin of BE.[8] Further, the trajectory connected intestinal metaplasia cells to the malignant cells, resembling the dysplastic developmental route of EAC (Figure 4B).

To understand the differences between the intestinal metaplasia cells and their non-metaplastic neighbors in the trajectory (chief cells and foveolar cells, called BE/fundus group in the following), we performed a differential expression analysis (DEA) followed by a GSEA. Among the highest-ranking genes with significantly increased expression in cells of the BE/fundus group were the pepsinogen coding genes *PGA3*, *PGA4*, and *PGA5* (Figure S9A; Table S6). Furthermore, *PGC* and *LIPF* were among these top differentially expressed genes, being markers for cells commonly found in gastric pits. On the side of intestinal metaplasia cells, the highest-ranking markers were *regenerating family member 4* (*REG4*), *TFF3*, *PHGR1*, and claudins, representing gastrointestinal and goblet cell-like features. When more specifically comparing foveolar-1 with intestinal metaplasia cells, we observed a downregulation of *CLDN18* and an upregulation of *CLDN3*, *CLDN4*, and *CLDN7* in intestinal metaplasia cells (Figures S9B–S9D; Table S6), in agreement with a shift toward a more metaplastic phenotype.[19,20]

An enrichment of gene set analysis further showed that terms related to gastric cells were among the upregulated processes in BE/fundus, while intestinal metaplasia cells showed enrichments in terms related to intestinal cells, suggesting a functional adaptation to a more intestinal phenotype. Moreover, intestinal metaplasia showed enrichments in terms related to oxidative phosphorylation and cell-cell interactions, supporting a metabolic shift and epithelial change. Intestinal metaplasia cells lacked detected SCNAs (see above) and had a higher metabolic turnover (oxidative phosphorylation), thereby connecting a hallmark of cancer with a non-cancer cell type.

## Enrichment of genetic risk loci in cell-type-specific gene expression profiles

We wanted to understand through which cell types and biological processes inherited genetic risk profiles can lead to the development of BE and EAC. Therefore, we performed a partitioned heritability analysis[21,22] where single-nucleotide polymorphisms (SNPs) were categorized based on their contribution to the heritability of BE and EAC, respectively. Based on the scRNA-seq data, we defined sets of genes that were specifically expressed in each cell type. Next, we used the method linkage disequilibrium (LD) score regression (LDSR) on our BE and EAC GWAS data[9] to test whether risk loci that are located in 100 kb regions surrounding the most specifically expressed genes of each cell type show an enrichment of BE and/or EAC associations. This analysis tests whether SNPs in cell-type-specific genes show a stronger BE/EAC association when partitioned for their LD context than all GWAS SNPs. In other words, we tested whether the genes that are typically expressed in a given cell type show an enrichment for GWAS-derived risk loci of BE and/or EAC. Of note, for this analysis, it is irrelevant whether the sequenced cells are derived from a patient with or without the respective risk variants, since generic cell-type transcriptomic profiles are analyzed and no tests for allelic associations are performed.

Overall, EAC and BE risk variants showed significant enrichments among the most specifically expressed genes (gene sets) from epithelial cells and some specific fibroblasts, endothelial cells, and immune cells (Figures 5A–5C; Table S7). This was in contrast to risk loci for major depressive disorder (MDD),[23] which served as a negative control. In addition, we used lung cancer risk loci as a control and did not observe significant enrichments (Table S8). The failure to find any enrichment of MDD and lung cancer associations validated the concept of this study. By focusing on single genes, we were able to prioritize candidate genes located within GWAS loci for EAC and BE (Tables S9 and S10, respectively).

In terms of significant enrichment, we observed a stronger fold enrichment of EAC (*n* = 10) over BE (*n* = 5) risk loci among 31 cell types (Figures 5A–5C). This implies that EAC development is driven to a greater extent by local cellular processes than the development of BE.

Intestinal metaplasia cells showed the strongest enrichment for EAC risk loci among nonmalignant cells, and the enrichment was substantially weaker for BE risk loci (Figure 5A). This suggests that biological processes within intestinal metaplasia cells play an important role in the malignant transformation into EAC through germline genetic risk. This adds to the evidence of a functional and developmental relationship between BE and EAC as these non-cancer cells are abundant in both BE and EAC tissue samples (Figure 2C). This is also reflected in our trajectory analysis, in which the intestinal metaplasia cell was the pivotal cell type in the transition from benign to cancer cells (Figure 4B).

We also performed additional analyses to assess whether the presence of cancer substantially alters the enrichment signal across cell populations. By comparing LDSR-derived coefficient

---

**Figure 4. Characterization of epithelial cells shows gastric features for intestinal metaplasia cells**

(A) Gene set enrichment analysis (GSEA) of epithelial clusters. The enrichment of gene sets from Busslinger et al.[16] with a relationship to BE or the esophagus was tested for enrichment in cell types (columns). The top row represents the origin of the clinical sample for each cell type.

(B) Trajectory analysis of epithelial cells. UMAP representation of epithelial cells as in Figure 2A superimposed with the pseudotime trajectory (black lines) as a proxy for relatedness.

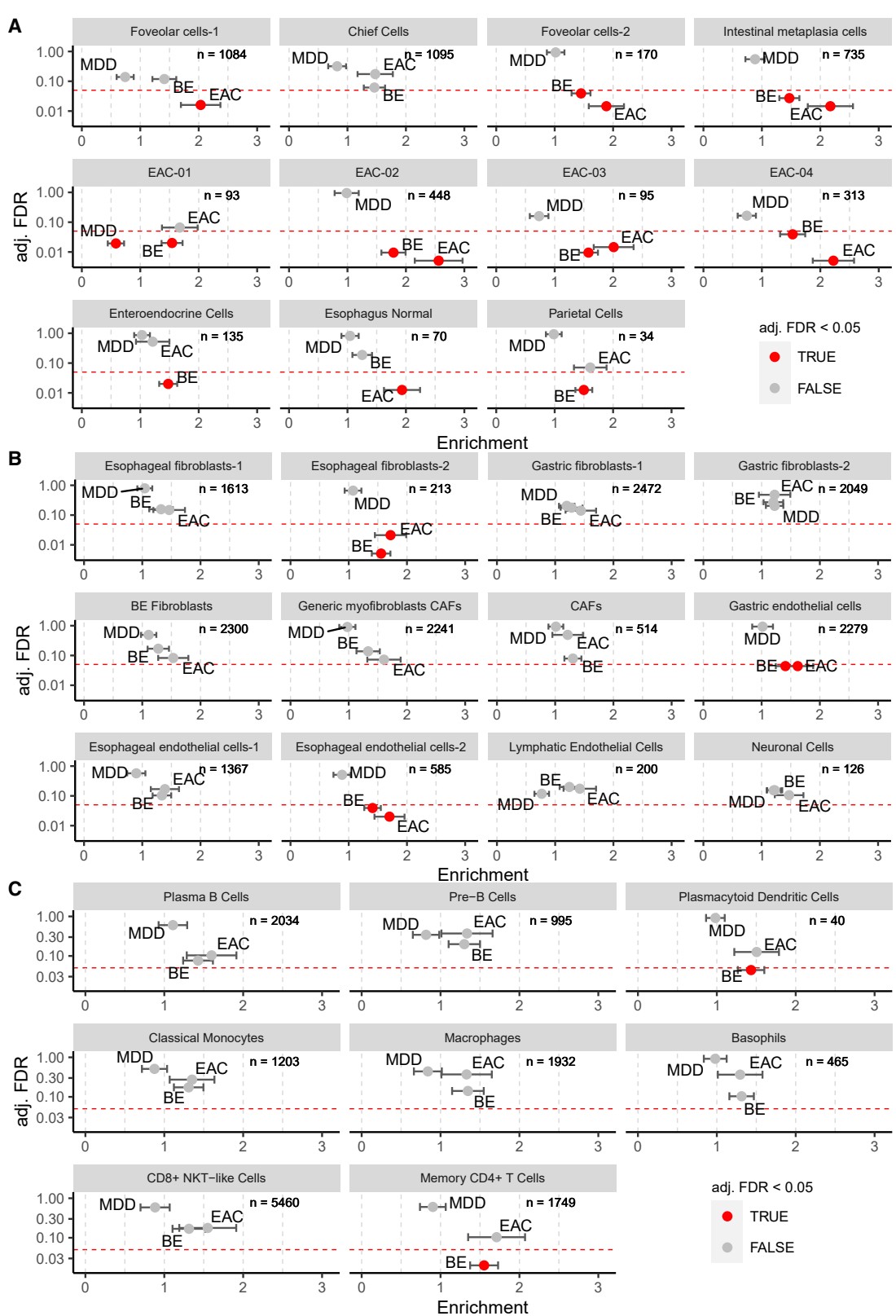

*(legend on next page)*

z-scores between matching and non-matching cell populations from BE and EAC patients, we found that matching cell populations exhibited significantly lower variance in enrichment signals ($p = 0.02$, F-test; $p = 0.03$, Levene's test, Figure S11). These results suggest that, despite possible tumor-related remodeling, shared cell populations retain relatively stable genetic enrichment patterns. Nonetheless, the tumor microenvironment may still contribute to shifts in gene regulation or cellular states, which is an important avenue for future studies.

Esophageal-fibroblasts-2, a cell type exclusively derived from normal esophagus, was enriched for EAC and BE risk loci, providing evidence that fibroblasts may contribute to the etiology of EAC and BE by germline genetic risk (Figure 5B). In addition, the endothelial cells of esophageal endothelial cells-2 and gastric endothelial cells, the majority of which were derived from normal esophagus and fundus, respectively, were significantly enriched for BE and EAC risk variants (Figure 5B), suggesting a role in disease formation. Further research is necessary to determine the role of this cell type in neoangiogenesis as a way of promoting tumor progression by nonmalignant cells. Among immune cells, plasmacytoid dendritic cells and CD4$^+$ memory T cells showed enrichment exclusively for genes at BE risk loci (Figure 5C), indicating that processes related to the innate as well as the adaptive immune system may contribute to BE development through germline genetic risk.

Of all the tumor cell clusters, EAC-02 showed the strongest enrichment, while EAC-01 showed no significant enrichment of EAC risk variants (Figure 5A). This suggests that EACs can emerge on the basis of different components of germline genetic risk.

Intestinal metaplasia cells characterize the transition into EAC (Figure 4B) and show substantially stronger enrichment for EAC than for BE risk loci (Figure 5A). These cells are only present in BE and EAC tissue (Figure 2C) and represent nonmalignant epithelium (Figure 4B). We, thus, used a second scRNA-seq dataset of BE epithelial cells from the study of Nowicki-Osuch et al.[8] in order to increase the cell-type resolution. Multiple different epithelial cell types could be identified in this study. Applying partitioned heritability and LDSR analysis on these data, we observed enrichment of BE and/or EAC risk loci among genes that characterize columnar cell types. Again, MDD risk loci that served as negative control showed no enrichment, and among significant enrichments, a stronger fold enrichment among cell types was observed in EAC ($n = 22$) compared to BE ($n = 3$, Figure 6) risk loci. While the enrichment was present for BE and EAC risk loci in columnar-undifferentiated, columnar-dividing, and columnar-intermediate cell types (Figure 6), the enrichment in the columnar-differentiated cell type was only present for EAC risk loci (Figure 6). These findings point to cell bio-

logical processes that control the differentiation of columnar cell types as potentially relevant for the transformation into EAC. Marker genes of columnar-differentiated cells showed the strongest enrichment in intestinal metaplasia cells among non-cancer epithelial cells of our study (Figure S12), suggesting their presence within intestinal metaplasia cells and supporting the importance of this cell type for genetic-risk-mediated EAC development. Marker genes for intestinal metaplasia cells overlapping with columnar-differentiated cells include *REG4*, *cadherin-17* (*CDH17*), *alanyl aminopeptidase formerly CD13* (*ANPEP*), *mucin 17* (*MUC17*), and *fatty acid binding protein 2* (*FABP2*).

Overall, the analysis suggests the contribution of several cell types to the development of BE and EAC and the stronger contribution of local cellular processes influenced by germline disposition for EAC compared to BE, with columnar-differentiated cells within intestinal metaplasia cells showing a strong contribution.

## DISCUSSION

In contrast to earlier studies on BE or EAC scRNA-seq,[7,8,16,24,25] our focus was to identify the cell types and biological processes that contribute to the development of BE and/or EAC through germline genetic risk (partitioned heritability[9,21,22]).

Although the BE sample size used in the GWAS is much larger compared to EAC (11,208 patients with BE versus 5,582 patients with EAC[9]) and thus has a higher statistical power to detect enrichment, EAC risk loci showed stronger enrichment among most of our epithelial cell types. Of all 31 cell types analyzed, 15 showed significant enrichment for EAC and/or BE risk loci. Of these, 10 showed stronger enrichment for EAC risk loci, whereas only 5 showed stronger enrichment for BE risk loci (Figure 5). This finding was validated in the independent scRNA-seq of Nowicki-Osuch et al.[8] Twenty-two of the analyzed cell types showed stronger enrichment for EAC than BE risk loci, and three cell types showed stronger enrichment for BE than EAC risk loci (Figure 6). Our results imply that EAC development is driven to a greater extent by local cellular processes than the development of BE. The finding confirms the results of our GWAS,[9] where it has been shown that non-local pathomechanisms, like GERD and hiatal hernia, contribute substantially more strongly to the metaplastic BE transformation than to EAC development.

Of all cell types, our data suggest that intestinal metaplasia cells are of utmost importance for the malignant transformation to EAC. Our trajectory analysis showed that intestinal metaplasia cells represent the non-cancer cell type that is closest to EAC (Figure 4B). Accordingly, intestinal metaplasia cells showed the strongest enrichment for EAC risk variants among all nonmalignant cells and, to a lesser extent, for BE risk loci (Figure 5).

**Figure 5. Partitioned heritability analysis of GWAS risk loci for BE and EAC applied on cell-type-specific expression profiles**
Fold enrichment of BE, EAC, and major depressive disorder (MDD; negative control) GWAS risk loci in cell-type-specific gene sets is shown on the *x* axis; adjusted (adj) false discovery rate (FDR) is shown on the *y* axis. Standard errors were estimated by jackknifing over blocks of individuals. Standard errors were used to correct for attenuation bias in LD Score regression.
(A) Epithelial cells.
(B) Fibroblasts, endothelial cells, esophagus normal cells, and neuronal cells.
(C) Immune cells. Enrichment describes the level of enrichment of GWAS risk variants within cell-type-specific gene sets. Significance level with an FDR of 0.05 is indicated by a red dashed line. Significant enrichments are indicated by red data points.

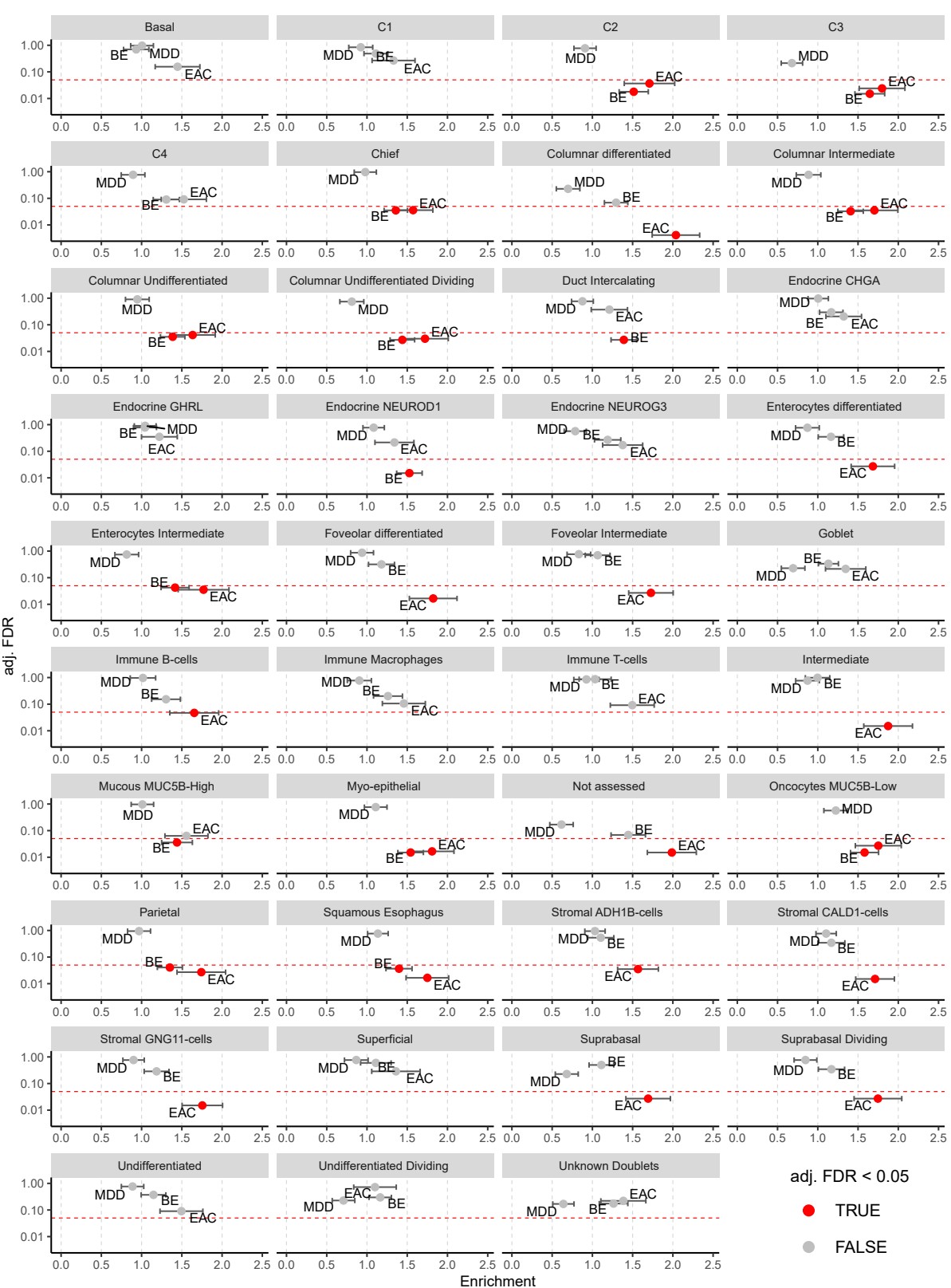

*(legend on next page)*

Intestinal metaplasia cells consist of nonmalignant cells derived from BE and EAC epithelium, and their benign state is in agreement with the absence of large-region SCNAs according to our copy-number profile analysis (Figure 3B).

Based on the impact of intestinal metaplasia cells on malignant transformation, we used the scRNA-seq data of Nowicki-Osuch et al.[8] and applied partitioned heritability/LDSR analyses. In their study, the BE epithelium was analyzed in more detail, which led to the identification of a diversity of different epithelial cell types. We found that columnar-undifferentiated, columnar-dividing, and columnar-intermediate cell types showed significant enrichment for BE and EAC risk loci. In contrast, the columnar-differentiated cell type exclusively showed enrichment for EAC risk loci (Figure 6). It is known that columnar epithelial cells develop from undifferentiated cells via intermediate states to differentiated cells. Among these cell types, the undifferentiated columnar cells constitute the origin of EAC according to Nowicki-Osuch et al.[8] Our data support the notion that differentiated and undifferentiated columnar cells are of relevance in EAC development. This is based on the transcriptomic enrichment of risk loci, where the expression profiles of differentiated columnar cells show the strongest enrichment. This is in line with the increased risk of BE (usually characterized by intestinal metaplasia) consisting of columnar cells to develop EAC.[1,2]

Genes that are characteristic of columnar-differentiated and intestinal metaplasia cells include *REG4*, *CDH17*, *ANPEP*, *MUC17*, and *FABP2*. *REG4* is a member of the calcium-dependent lectin gene family. It is a marker for deep crypt secretory cells of the small and large intestine that provide secretory support for stem cells at the bottom of crypts.[26] Upregulation of *REG4* has been reported for gastrointestinal cancers, where its dysregulation has been connected with proliferation, invasion, and drug resistance.[27,28] *REG4* promotes invasion, proliferation, and tumor growth but also the migration of gastric cancer cells.[29,30] *CDH17* is a marker of differentiation in intestinal cells and promotes cell adhesion and proliferation in colorectal cancer.[31] *ANPEP* is a surface marker of mature zymogenic chief cells in the gastric epithelium[32] and is downregulated in colorectal cancer compared to normal colon tissue.[33] *MUC17* encodes for a transmembrane glycoprotein with a large extracellular domain. It is expressed in intestinal cells[34] and assumed to protect the intestinal epithelium from microbial invasion. *FABP2* is involved in the uptake and trafficking of lipids in the intestine.[35] It is primarily expressed in the tips of intestinal villi.[36] These markers describe a differentiated columnar cell type that can be found in BE and EAC tissue and likely contributes to EAC development through germline genetic risk variants. *OLFM4* characterizes undifferentiated columnar cells in BE, which were implicated by Nowicki-Osuch et al. as the cell of origin of EAC.[8] Here, we saw a strong expression of *OLFM4* in intestinal metaplasia cells, indicating that the differentiated and undifferentiated states of columnar cells in BE might be very

close. *OLFM4* is mainly expressed in the digestive system as a marker for intestinal stem cells and is upregulated in several types of cancers, including gastric cancer.[12,13,37] Knockdown of *OLFM4* promotes the migration of gastric cancer cells through NF-κB, and low expression is associated with lymph node metastases in early gastric cancer.[38]

Apart from the relevance of intestinal metaplasia and columnar epithelial cells to EAC development, we found that germline genetic risk factors also seem to influence BE and EAC through other cell types (Figure 5). This implies that a variety of different cellular mechanisms orchestrate BE and EAC development. A defined population of fibroblasts, namely esophageal fibroblasts-2, showed enrichment for BE and EAC risk loci. Whereas the role of CAFs in treatment response and the formation of metastases is well documented,[39] the influence of normal fibroblasts on the development of cancer is largely unknown and provides a new area for research. In addition, two endothelial cell types, esophageal endothelial cells-2 and gastric endothelial cells, were significantly enriched for BE and EAC risk loci, suggesting that blood vessels might be relevant for disease formation, presumably through neoangiogenesis. Further functional experiments and studies with collections of BE tissue of progressing/non-progressing patients are necessary to elucidate this relationship and to investigate the prognostic value of individual cell-type characteristics.

The distinct enrichment of BE risk loci in plasmacytoid dendritic cells and in memory CD4+ T cells further emphasizes the complexity of BE etiology. It is well known that the adaptive immune system is relevant for the elimination of (pre-)malignant cells carrying neoantigens. Our data thus suggest that germline genetic factors might influence the adaptive immune system and thus impact the immune-escape rate of mutation-carrying BE clones to eventually grow. Since it has been shown that BE already contains cancer-like mutations,[40] it is plausible that neo-epitopes at this benign stage can already be targets for immune response, involving memory T cells.

Finally, the partitioned heritability analysis showed that tumor cells are enriched for EAC risk loci to a different, tumor-dependent degree, ranging from high (EAC-02 and EAC-04) to low (EAC-01) enrichment. This finding is quite interesting and indicates that germline genetic factors are of different relevance for the development of individual EACs.

Overall, our study provides evidence that certain local cell types contribute to BE and EAC development through germline predisposition. These effects are substantially stronger in the development of EAC than BE, where non-local factors such as GERD and hiatal hernias are also involved. The data suggest that intestinal metaplasia cells are a cell type within BE of pivotal relevance for EAC development. Here, cell biological processes that control the differentiation of columnar epithelial cells seem to be involved in EAC development. Thus, the diagnostic use of markers that characterize the differentiation of columnar

**Figure 6. Partitioned heritability analysis of GWAS risk loci for BE and EAC applied on gastroesophageal junction cell-type-specific expression profiles of Nowicki-Osuch et al.**
See Figure 4 for a description. C1, C2, C3, and C4: KRT7high (supra)basal, submucosal gland-like cells; duct: submucosal duct; CHGA, GHRL, NEUROD1, NEUROG3, MUC5B, ADH1B, CALD1, and GNG11: marker genes for cell types of Nowicki-Osuch et al.[8]

epithelial cells should be explored in the future for EAC prediction.

## Limitations of the study

Sample acquisition remains the biggest challenge in scRNA-seq experiments. As in our study, sample sizes are often limited due to the time- and resource-consuming processing of tissue samples. The nature of EAC progression from BE makes it difficult to establish datasets of progressors and true non-progressors. The observational data imply many promising cell-type interactions and their functional relevance. Further research is needed to confirm our findings on a functional level.

## RESOURCE AVAILABILITY

### Lead contact

Further information and requests for reagents may be directed to and will be fulfilled by the lead contact, Axel M. Hillmer (ahillmer@uni-koeln.de).

### Materials availability

This study did not generate new unique reagents.

### Data and code availability

- The scRNA-seq data have been deposited at the European Genome-Phenome Archive (EGA) with accession number EGAS50000000530.
- GWAS data were previously published,[9] with the dataset for BEACON available in dbGAP with accession number: phs000869.v1.p1, the dataset of the Cambridge cohort available via GWAS Catalog with accession number: GCST003738, and the PGC consortium with accession number: 29700475.
- The scRNA-seq data of Nowicki-Osuch and colleagues are available from the European Genome-phenome Archive (EGA), www.esophaguscancercellatlas.org, and the HCA project, as indicated in Nowicki-Osuch et al.[8]

## ACKNOWLEDGMENTS

The study was supported by the Wilhelm Sander Foundation with grant 2020.119.1; the German Research Foundation (DFG) with grants 418074181, INST 216/1063-1 FUGG grant 446411360, and CRC 1310 C4; and the Federal Ministry of Education and Research (BMBF) with grant CompL DeepInsight 031L0267B. Figure 1A and the graphical abstract were created with BioRender.com. We thank the patients who participated in the study. The R.C.F. lab is supported by the Medical Research Council (MR/W014122/1).

## AUTHOR CONTRIBUTIONS

J.S. and A.M.H. designed and coordinated the study. P.S.P., S.H., C.J., C.J.B., Y.Z., and S.-H.C. collected clinical samples. P.S.P., S.H., R.B., A.Q., and S.-H.C. obtained and analyzed clinical data. P.S.P. and S.H. dissociated tissue samples. M.F. performed NGS. M.C.W. and A.-S.G. performed bioinformatics analyses. M.C.W., A.-S.G., P.D., D.H., C.P., and C.M. performed data analysis. M.C.W., A.-S.G., R.T., Y.Z., R.C.F., R.B., A.Q., I.G., C.M., J.S., and A.M.H. interpreted the data. M.C.W., J.S., and A.M.H. drafted the manuscript.

## DECLARATION OF INTERESTS

The authors declare no competing interests.

## STAR★METHODS

Detailed methods are provided in the online version of this paper and include the following:

- KEY RESOURCES TABLE
- EXPERIMENTAL MODEL AND STUDY PARTICIPANT DETAILS
  - Ethics approval and consent to participate
  - Patient cohort
  - Sample processing
- METHOD DETAILS AND QUANTIFICATION AND STATISTICAL ANALYSIS
  - Sequencing and raw data processing
  - Quality control of the single cell RNA-sequencing data
  - Batch correction, normalization, clustering, and visualization
  - Differential expression analysis and cell type annotation
  - Cell-cycle scoring
  - Gene set enrichment analysis
  - Inferring somatic copy number variations with *inferCNV*
  - Trajectory analysis using *Monocle3*
  - Partitioned heritability
  - Satistics and reproducibility
  - More details are below
  - Identification of epithelial cell types

## SUPPLEMENTAL INFORMATION

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

## STAR★METHODS

### KEY RESOURCES TABLE

| REAGENT or RESOURCE | SOURCE | IDENTIFIER |
| --- | --- | --- |
| **Antibodies** | | |
| CD45, PE/Cy7-conjugated antibodies | Biolegend | Order nr. 304015; RRID: AB_314403 |
| TotalSeq[TM]-B against CD45 | Biolegend | Clone HI30, order nr: 304066; RRID: AB_2800761, barcode sequence TGCAATTACCCGGAT |
| TotalSeq[TM]-B against CD90 | Biolegend | Clone 5E10, order nr: 328147; RRID: AB_2832642, barcode GCATTGTACGATTCA |
| TotalSeq[TM]-B against CD326 | Biolegend | Clone 9C4, order nr: 324249; RRID: AB_2860852, barcode TTCCGAGCAAGTATC |
| TotalSeq[TM]-B against CD31 | Biolegend | Clone WM59, order nr: 303145; RRID: AB_2832582, barcode ACCTTTATGCCACGG |
| **Biological samples** | | |
| Patients will be or EAC | University Hospital Cologne | institutional review board of the University of Cologne: 18-274 |
| **Chemicals, peptides, and recombinant proteins** | | |
| DNAse I | AppliChem PanReac | Order nr. A3778 |
| Collagenase IV | Thermo Fisher Scientific | Order nr. 17104019 |
| Dispase II | Sigma-Aldrich | Order nr. D4693-1G |
| Fetal bovine serum (FBS) | Capricorn Scientific | Order nr. AN-22643 |
| Propidium iodide | Thermo Fisher Scientific | Order nr. P1304MP |
| Human TruStain FcX[TM] | BioLegends | Order nr. 422301 |
| Hoechst | Thermo Fisher Scientific | Order nr. H1399 |
| 3′ v3.1-paired end chemistry | Illumina | NovaSeq 6000 S2 Reagent Kit v1.5 (100 cycles) Order nr. 20028316 |
| **Critical commercial assays** | | |
| CITE-Seq | 10x Genomics | 3′ Feature Barcode Kit, 16 rxns Order nr. PN- 1000262 |
| Chromium Single Cell 3′ Solution | 10x Genomics | Chromium Next GEM Single Cell 3' Kit v3.1, 16 rxns Order nr. PN-1000268 |
| **Deposited data** | | |
| scRNA-seq data generated as part of this study | European Genome-Phenome Archive (EGA) | EGAS50000000530 |
| GWAS data of EAC and BE: a) BEACON (Barrett's and Esophageal Adenocarcinoma Consortium) being available in dbGAP; b) Cambridge cohort being available via GWAS catalog | a) dbGaP; b) GWAS Catalog | a) dbGaP: phs000869.v1.p1, Data access: https://dbgap.ncbi.nlm.nih.gov/)eta/search/?OBJ=study&TERM=BEACON, b) GWAS CatalogAccession number: GCST003738, Data access: https://www.ebi.ac.uk/gwas/studies/GCST003738; Schroder et al.[9] |
| GWAS data of MDD | PGC consortium | PGC consortium: 29700475; Data access: https://pgc.unc.edu/for-researchers/download-results/; Wray et al.[23] |
| scRNA-seq data of BE | Nowicki-Osuch et al.[8] | |
| MSigDB | UCSanDiego/Broad Institute | https://www.gsea-msigdb.org/gsea/msigdb |
| **Software and algorithms** | | |
| Cell Ranger v6.1.2 | 10x Genomics | Zheng et al.[41], RRID:SCR_017344 |
| Seurat V4 | Satija Lab | Hao et al.[42], RRID:SCR_016341 |
| R | | https://www.r-project.org/ |
| R package *fgsea* for gene set enrichment analysis | | Korotkevich et al. bioRxiv. https://doi.org/10.1101/060012 |
| R package *inferCNV* | | Tirosh et al.[10], RRID:SCR_021140 |

*(Continued on next page)*

*Continued*

| REAGENT or RESOURCE | SOURCE | IDENTIFIER |
|---|---|---|
| R package *Monocle3* | | Cao et al.[43], RRID:SCR_018685 |
| LDSR partitioned heritability method | | Finucane et al.[22] |
| MAST | | Finak et al.[44], RRID:SCR_016340 |
| Other | | |
| GentleMACS Dissociator | Miletnyi Biotec | Order nr. 130-093-235, RRID:SCR_020267 |
| Flow Sorter LE-MA900FP | Sony | LE-MA900FP, RRID:SCR_026300 |
| Chromium platform | 10x Genomics | Chromium X (RRID:SCR_024537) & Accessory Kit PN-1000331 |
| NovaSeq 6000 | Illumina | Order nr. 20012850, RRID:SCR_016387 |

## EXPERIMENTAL MODEL AND STUDY PARTICIPANT DETAILS

### Ethics approval and consent to participate

The study was approved by the institutional review board of the University of Cologne (18–274), follows the principles of the convention of Helsinki, and informed consent was obtained from all participating patients.

### Patient cohort

Samples from two groups of patients were analyzed, one comprising five patients with pre-malignant metaplastic BE and one with four EAC patients. All patients were males of European descent. During diagnostic esophagogastroscopy biopsies were taken from the metaplastic or cancer tissue along with biopsies from normal esophageal epithelium (>5 cm proximal of BE/EAC) and normal gastric samples from the fundus.

### Sample processing

Fresh tissue samples from endoscopies, taken either from the site of the pathologies or at a distance of 5 cm apart for normal tissue, were dissociated as described earlier.[45] In brief, the tissue was minced to small pieces, disrupted in a C-tube used with the GentleMACS Dissociator (Miletnyi Biotec) combined with enzymatic digestion with DNAse I (500 $U \cdot mL^{-1}$; AppliChem PanReac), collagenase IV (320 $U \cdot mL^{-1}$; Thermo Fisher Scientific), and dispase II (2 $U \cdot mL^{-1}$; Sigma-Aldrich), filtered through a 100 $\mu m$ cell strainer, collected and resuspended in 60% RPMI-1640 medium (Thermo Fisher Scientific), 30% FBS (Capricorn Scientific), and 10% dimethyl sulfoxide (DMSO) (Sigma-Aldrich) for freezing at $-80°C$. After 24 h samples were transferred to liquid nitrogen until fluorescence-activated cell sorting (FACS). FACS was conducted prior to sequencing using MA900-FP (Sony).

The samples from patients will be were processed in the following way: Single-cell suspensions were stained with PE/Cy7-conjugated antibodies against CD45 (Biolegend) as a leukocyte marker and with propidium iodide (Thermo Fisher Scientific) to distinguish live and dead cells according to the manufacturer's specifications. For the cells from EAC samples, dissociated single-cell solutions were prepared for Cellular Indexing of Transcriptomes and Epitopes by Sequencing (CITE-Seq) analysis. After thawing, cells were prepared with the kit for the cell surface protein labeling for single cell RNA sequencing (10x-Genomics) according to the manufacturer's instructions. First, cells were labeled with 2 µL preconjugated antibodies per 100 µL total volume. Antibodies (category: TotalSeq-B) against CD45 (order nr: 304066, barcode sequence TGCAATTACCCGGAT), CD90 (order nr: 328147, barcode GCATTGTACGATTCA), CD326 (order nr: 324249, barcode TTCCGAGCAAGTA), and CD31 (order nr: 303145, barcode ACCTTTATGCCACGG) were purchased from BioLegends and each labeled with specific barcode sequences. To avoid unspecific binding, Human TruStain FcX (Fc Receptor Blocking Solution) (BioLegends) (5 µL per 100 µL total volume) was added. In the next step, cells were washed to eradicate excessive antibodies. Afterward, single-cell suspensions were stained with Hoechst (staining of all cells) and propidium iodide (staining of dead cells) (Thermo Fisher Scientific) to distinguish live and dead cells according to the manufacturer's specifications. FACS was performed using a 100-$\mu m$ nozzle on MA900-FP (Sony). For each sample, 10,000 living cells were sorted. For samples with excessive leukocyte presence, CD45-positive cells were restricted to maximum 25% of total cells. Collected single cells were placed on ice and further processed by the Cologne Center for Genomics (CCG, Cologne) for 3′ single-cell RNA-sequencing with the 10x Genomics' scRNA-seq technology (Chromium Single Cell 3′ Solution) according to the manufacturer's specifications. Targeted cell recovery was aiming at 3,000 cells per sample.

## METHOD DETAILS AND QUANTIFICATION AND STATISTICAL ANALYSIS

### Sequencing and raw data processing

The samples were processed using the 10x Genomics Chromium platform and sequenced on a NovaSeq 6000 (Illumina) with Illumina 3′ v3.1-paired end chemistry, 29-10-10-89 bp aiming at 25,000 read pairs per cell. Subsequently, the raw data was processed using

the 10x Genomics software Cell Ranger v6.1.2.[41] Seurat V4 was then used for further handling of the data.[42] Procedure of quality control, batch correction, and normalization are described below.

### Quality control of the single cell RNA-sequencing data

The output matrices of Cell Ranger were loaded into Seurat using R. Cells with at least 1,000 different sequenced genes were retained from BE patients. For cells from EAC patients and patient BE-pat02 the threshold was lowered to 500 to capture a wider spectrum of (tumor) cells. The quality of cells was inspected regarding the total number of counts per cell, the number of different genes, and the fraction of mitochondrial genes. Three median absolute deviations were allowed for the first two parameters to account for very low-quality cells and to reduce the chance to capture doublets. For the fraction of mitochondrial genes, a rather high initial threshold of 30% was chosen allowing cells with naturally higher fractions of mitochondrial mRNA to be retained. After initial clustering, the top 15% of the cells per cell type regarding that parameter were discarded to set a cell type dependent threshold.

### Batch correction, normalization, clustering, and visualization

To account for technical artifacts due to processing of cells in different batches the Canonical Correlation Analysis method included natively into the Seurat package was used before clustering and visualization.[46] True batches, i.e., samples that were sequenced together, were merged without using a batch correction method. The data were normalized using Seurats "LogNormalize" function. Using the function *FindVariableFeatures,* 6,000 features were identified using the 'vst' option. Principal component (PC) analysis was performed and the number of retained PCs was determined by the point at which the smoothed difference between the accounted variability of two adjacent points was lower than 0.02 using a rolling average of two additional points in both directions. To find clusters the recommended functions *FindNeighbors* using a KNN graph based on the euclidean distance in PCA space and analyzing the Jaccard similarity and *FindClusters* which employs the Louvain algorithm were used.[42] The *UMAP* algorithm was used to find a two-dimensional projection of the data.

### Differential expression analysis and cell type annotation

*FindAllMarkers* was used to find cluster-specific marker genes. Therefore, a minimal percentage of cells of 30% expressing a tested gene within a given cluster and a minimal logFC threshold of 0.1 were chosen as parameters. To minimize putative batch effects, we included the batch as a latent variable into the model. *MAST* was incorporated in the model by choosing the respective option.[44] To the results, the ratio between the fraction of cells within a given cluster expressing a given gene and the fraction in all other clusters was calculated to find a measure of cluster-specificity (pct.ratio). As marker genes, the genes that had the highest logFC between clusters and the highest pct.ratio were considered. Typical markers were found for the initial cell type annotation. *PanglaoDB* was used for a semi-automatic way of cell type annotation.[47] The immune cells were annotated following the standard workflow of *ScType*.[48]

### Cell-cycle scoring

Using the method of Tirosh et al.,[10] a correlation between the expression profiles of single cells and cell cycle expression signatures were calculated to assign the most probable cell cycle phase out of G1, S and G2/M. The number of cells per cluster being assigned to the S and G2/M phase were divided by the number of G1 cells in that cluster to assign the relative proliferation index.

### Gene set enrichment analysis

Gene set enrichment analysis (GSEA) was performed on gene lists obtained by the Seurat functions *FindAllMarkers* or *FindMarkers* using a logFC threshold of 0.05 and a minimum percentage of cells expressing a given gene in the tested cluster of 0.3. The used model was *MAST*. The R package *fgsea* was used to run the analysis (Korotkevich et al. bioRxiv. https://doi.org/10.1101/060012) using the databases of MSigDB.[7–9]

### Inferring somatic copy number variations with *inferCNV*

The R package *inferCNV*[10] was used to estimate collective SCNAs using a sliding window of genes, which was set to 150. The algorithm explores expression levels of genes across the genome in comparison to a reference set of cells. The settings as recommended were used to generate the results. The order of genes follows their physical position on a given chromosome even though zero-expression genes have been discarded. The approximate physical position of the centromere for each chromosome is marked at the last included gene physically positioned before CENP-A of the respective chromosome.[49] If that particular gene had been discarded due to zero expression, the next included gene toward the end of the chromosome was considered as a proxy.

### Trajectory analysis using *Monocle3*

For trajectory analysis we used the R package *Monocle3*, using the option close_loop = TRUE.[43]

### Partitioned heritability

Gene-sets derived from the scRNA seq data were used to create genome annotations. Gene-sets for each of the 31 identified cell types or clusters were constructed as follows. All genes were excluded that showed no expression in any cell type or cluster. Next, a

gene's cell type or cluster expression specificity was calculated by dividing the expression of each gene in each cell type or cluster by the total expression of this gene in all cell types and clusters. Thus, the specificity ranges between 0 and 1, where 1 means that a gene is specifically expressed in a cell type or cluster, while 0 implies no expression in the respective cell type or cluster. For each cell type or cluster, the top 15% most specific genes were used to construct gene-sets. The specificity metric follows the approach of Bryois et al.[21] Next, we used the LDSR partitioned heritability method[22] and GWAS data on BE and EAC[9] as well as MDD.[23] Here, we tested whether GWAS SNPs that are located in 100kb regions surrounding the most specifically expressed genes of each gene-set show an enrichment of BE and/or EAC as well as MDD associations. In order to correct for multiple testing, we applied an FDR-adjustment on the enrichment results.

## Satistics and reproducibility

Seurat V4 with the implemented statistical analysis in R was used for definition of single cell RNA-seq clusters.[42] The differential gene expression analysis was performed using MAST[44] where pairs of regression coefficients through bootstrapping and an empirical Bayesian framework are used. LDSR partitioned heritability method was applied as described by Finucane and colleagues.[22]

## More details are below
### Mesenchymal and immune cells

To identify mesenchymal cells, we used the markers *VWF*, *PECAM1*, *PLVAP* and *CD34* for endothelial cells, *DCN*, *DPT*, *LUM* and *PDGFRA* for fibroblasts and *RGS5*, *TAGLN*, *ACTA2* and *MYL9* for myofibroblasts. Cluster 17 had a specific expression pattern and its top markers could be linked to lymphatic endothelial cells using *PanglaoDB*.[47] Immune cells were identified by the expression of *CD2* and the *CD3* subunits *CD3D* and *CD3E* for T cells, *CD79A* for all B cells and *FPR1*, *S100A8* and *S100A9* for monocytes. Cluster 20 had a specific expression of *NRXN1* and *PLP1*, identifying the origin of these cells as neuronal (for cluster nomenclature see Figures S2A and S2C).

Fibroblasts, endothelial cells and immune cells were subjected to subanalysis, respectively. This revealed different fibroblastic states (Figure S5A). A myofibroblast-like population could be discerned by the expression of *ACTA2* and a classical fibroblast-like population was marked by the expression of *POSTN*. Among the myofibroblast-like population, a cluster annotated as Generic Myofibroblasts/CAFs (cancer-associated fibroblasts) that consisted of cells from all sampled tissue origins, suggesting a cell type state that remains stable throughout the course from normal tissue via BE to EAC (Figure S5B). A group of cells with high *ACTA2* expression almost exclusively contained cells from the cancer samples, which was therefore annotated as CAFs and a cluster annotated as esophageal fibroblasts-2 with cells from normal esophageal tissue. The clusters that expressed *POSTN* were homogeneous regarding sample origin (gastric fibroblasts-1, gastric fibroblasts-2, BE-fibroblasts, esophageal fibroblasts-1 and esophageal fibroblasts-2) but consisted of different batches, suggesting that the distinct clusters did not form based on technical artifacts but rather represent genuine cell type states with differences due to the tissue origin (Figure S5C). We compared the fibroblast cell populations with high *ACTA2* expression and high *POSTN* expression (Figure S5D). Typical marker genes discerned the two groups, with *RGS5*, *TAGLN*, *THY1* and *FABP4* as additional typical marker for a myofibroblast-like phenotype and *DPT*, *LUM*, *DCN* and *PDGFRA* as markers for the classical fibroblast-like group. GSEA of the genes differentially expressed in the *ACTA high*-group illustrated the typical functions of myofibroblasts such as *Muscle Contraction*, *Oxidative Phosphorylation* and *Cytoskeleton Organization* but not *Cell Migration* and manipulation of the ECM while the *POSTN high*-group is characterized by modulation of metabolic and immune process (Figure S6A).

Among the most differentially expressed genes between the generic myofibroblast-like cluster and the CAFs were genes such as *LUM*, *VCAN*, *MMP2*, *FN1* and *TIMP1* that are widely accepted as markers for CAFs. Running GSEA on the differentially expressed genes discerning these two groups, CAFs showed enrichments in *Epithelial-to-Mesenchymal Transition*, the *Matrisome* and a *Multi-cancer Invasiveness Signature* (data not shown).

Having neither elevated expression levels of *ACTA2* nor *POSTN*, esophagal fibroblasts-1 had special characteristics, as illustrated by GSEA of the differences between that group and the *POSTN high*-group. Functions included the *Homeostasis of Metal Ions*, *Proteolysis*, *Regulation of Immune Response* and *Secretion*, while the other group had a myofibroblast-like phenotype (Figure S6B).

Clustering of endothelial cells suggests tissue of origin specific characteristics with gastric fundus-derived endothelial cells separating from the rest, while clustering of immune cells was dominated by immune cell types. The immune cells were annotated using *ScType*, a tool for automated annotation of cell types, which works well for the already thoroughly characterized compartment of immune cells (Figures S6C and S6D).

In summary, the phenotype of fibroblasts is strongly influenced by the tissue of origin even across the related tissue environments of esophageal, gastric, Barrett's mucosa, and EAC. Myofibroblasts were characterized by typical muscle and cytoskeletal features, being separated from CAFs demonstrating features such as regulation of the ECM, promoting EMT and invasiveness in cancers. Classical fibroblasts showed enrichments in metabolic and immune processes separating them from the aforementioned clusters.

## Identification of epithelial cell types

We annotated the clusters chief cells, parietal cells, esophagus normal and enteroendocrine cells based on the expression of marker genes (Figure 3A). Chief cells expressed *PGA3*, *PGA4* and *PGA5*, encoding pepsinogen which is mostly secreted by gastric chief cells,[16] further supported by the high enrichment of the chief cell signature from Busslinger et al.[16] (Figure 4A). *ATP4A* and *ATP4B*

 CellPress

Cell Genomics
Article

encode for the ATP-dependent H + -pumps specific for parietal cells,[1] *KRT13* and *KRT15* have been reported to be specific for cells from the epithelium of the healthy esophagus,[16] and enteroendocrine cells were identified by the expression of *CHGA* and *GHRL*.[16,50]

Among the cancer cell clusters, *MMP7*, strongly expressed in EAC-01 (Figure 3A), is a matrix metalloproteinase that has been reported to have various oncogenic functions, such as inhibition of cancer cell apoptosis, reduction of cell adhesion and induction of angiogeneses (as reviewed by[51]). Garalla et al.[52] found *MMP7* to have peak expression in the invasive front of EAC compared to healthy and dysplastic tissue of the esophagus. *WNT11*, a characteristic gene for EAC-02 (Figure 3A), promotes non-canonical Wnt signaling pathway and has been involved in various cancers, including colorectal and gastric cancer as reviewed by Katoh[53] and Chen et al.[54] influencing tumor invasion and metastasis.[55]

The cluster parietal cells presented a high enrichment of the respective gene set from Busslinger et al.,[16] esophagus normal cells were enriched in the gene sets regarding esophageal late suprabasal cells, and both proliferating and quiescent basal cells of the esophagus. Busslinger et al.[16] reported the characteristics of gastric *D*, *G* and *X* cells in the context of enteroendocrine cells which showed a singular enrichment in the here found group of enteroendocrine cells.

