## [Document S2. Transparent peer review records for Wenzel et al · Cell Genomics]

Single cell analysis of Barrett's esophagus and carcinoma reveals cell types conferring risk via genetic predisposition

Author list: Marten C. Wenzel, Pouria Dasmeh, Patrick S. Plum, Ann-Sophie Giel, Sascha Hoppe, Marek Franitza, Christoph Jonas, René Thieme, Yue Zhao, Dominik Heider, Claire Palles, Rebecca Claire Fitzgerald, Christiane J. Bruns, Reinhard Buettner, Alexander Quaas, Ines Gockel, Carlo Maj, Seung-Hun Chon, Johannes Schumacher, Axel M. Hillmer

Summary

Initial submission: Received : Nov 12th 2024

Scientific editor: Judith Nicholson

First round of review: Number of reviewers: 2
Revision invited : January 31st 2025
Revision received : May 26th 2025

Second round of review: Number of reviewers: 1
Accepted : 29th July 2025

Data freely available: Yes

Code freely available: Yes

This transparent peer review record is not systematically proofread, type-set, or edited. Special characters, formatting, and equations may fail to render properly. Standard procedural text within the editor's letters has been deleted for the sake of brevity, but all official correspondence specific to the manuscript has been preserved.

Referees' reports, first round of review

Reviewer 1:

Wenzel et al have performed scRNAseq on a series of BE and EAC patients with samples from the stomach, esophagus, and BE/EAC. Combining this data with previously identified GWAS risk loci, the authors suggest specific cell types that may be influencing BE formation or progression to EAC. The idea to link GWAS risk loci to nearby genes and determine which cells express those genes is an interesting idea and the authors findings point to both epithelial and stromal/inflammatory cells contributing to BE formation and progression. While maybe not completely novel, the idea of non-epithelial cells contributing to progression is definitely a newer idea and further refinement of which epithelial cells are major contributors is also an area of ongoing investigation. Overall, further refining cells that influence BE formation and progression is an important area. With the limited number of patients that can be analyzed through scRNAseq, this however seems more like a starting point than the definitive results the authors put forth in the conclusions. The inclusion of a second scRNAseq dataset definitely helps, however, it seems larger validation or functional studies to validate the findings would help solidify the findings. Without this, a portion of the discussion should be given to the limitations of the study.

- 1) Could the findings be validated in a larger cohort of BE patients with known progression status utilizing a cheaper/high throughput technique like IHC/IF?
- 2) Overall the authors findings are interesting, however, I am not sure they fully support all of the conclusions. Without functional evidence, the authors need to be very careful on wording with interpretations. Some of their data is quite suggestive, but at least in my opinion, does not provide conclusive evidence of cause and effect. Many of the conclusions appear overstated and in my opinion, suffer from confounding associations with cause and effect. For example, statements such as "specific types of fibroblasts and endothelial cells contribute to BE and EAC development, while dendritic cells and CD4+ memory T cells contribute exclusively to BE development (abstract)", seem exaggerated without providing any functional evidence or at least any refuting alternative

explanations. There are many factors that can influence BE progression, it seems the methods used by the authors do a reasonable job at suggesting cell types influencing certain processes but it is not as clear that a negative result (no enrichment) means the cells are not contributing to BE progression or formation.

3) As above, The statement (line 281) "that germline genetic risk factors that control all steps of the cellular differentiation of the columnar BE epithelium contribute to EAC. This also includes the last step of columnar differentiation into differentiated cells that showed the strongest enrichment of EAC GWAS risk loci in the BE epithelium scRNA Seq data set." needs reworded or further explained. 1) It is confusing as written and 2) from what I understand it to mean, it is too strong of a statement. Could it be possible that the effect of the loci/expressed gene occurs early, but just happens to have sustained or increased expression in differentiated cells? Again, without functional evidence these statements seem exaggerated.

4) While I am not an expert in this area, I would assume GWAS hits for MDD would effect genes expressed in nervous tissue/brain. Given the very different tissue type, it does not seem unreasonable none of the loci are near genes expressed in the stomach/esophagus. What happens if GWAS hits for a different cancer that is closer in tissue type but doesn't show overlap in GWAS loci is used (?maybe lung or breast adenocarcinomas) as a control?

5) The GSEA (ln 129-135) for EAC-01/03 vs EAC-02/04 had upregulation of gene sets for squamous esophageal cells (suprbasal cells). Could this be contamination of normal squamous esophgeal cells in the cancer samples?

6) This may be my lack of experience with LDSR, but how would a risk allele that decreases expression of a gene (for example a risk allele that decreases expression of a tumor suppressor pathway) factor into the analysis? Similarly, if a risk loci is linked to a gene that is widely expressed by multiple cell types will this not show up in the analysis?

7) Line 215: "pointing at blood vessels influencing disease formation in terms of neoangiogenesis" This statement may be a little strong. What evidence is there this is due to neoangiogenesis, especially if the cell type was predominately found in normal tissue and not in regions where the BE or EAC develop?

8) If the patients with and without cancer were separated, does the non-epithelial cells (fibroblasts, endothelial cells,...) still show enrichment for EAC GWAS risk variants? If the enrichment holds for the BE patients, given that the majority of patients with BE will never progress to cancer, how does this affect the interpretation of the EAC GWAS hit findings?

9) In the opposite direction; Once an invasive cancer forms, even if small, it can exert an influence on the expression pattern and types of surrounding cells. This could affect expression profiles of both fibroblasts, endothelial cells, inflammatory cells, and even nearby non-neoplastic epithelial cells. If the EAC patients are contributing significantly to any of the findings, this seems like a potential caveat to the conclusions?

10) From looking at the marker expression in Figure 3 and as stated by the authors, it appears the BE_GFN cluster would resemble non-intestinal (foveolar-like) metaplasia and the BE_EAC cluster would resemble intestinal metaplasia (true BE). Given the known differences in risk of progression associated with non-intestinal vs intestinal metaplasia and the enrichment of BE_EAC in patients with EAC in this cohort, could this be confounding some of the conclusions?

1) (Minor) Though to a lesser extent, BE can contain CNVs that are also seen in EAC. Were these clusters completely copy neutral? Can the authors define "large scale copy gains". By large scale do they mean covering a large genomic region or many copy gains (high copy gain) in a specific region?

2) (Minor): For Sup Fig 6A, can the authors please explain what cells are in the BE/EAC transition zone besides the BE_EAC cluster? The results between BE/EAC and BE_EAC are slightly different, but it is hard to tell what other cells are in that circled area besides BE_EAC.

3) (Minor) Were the BE patients with exclusively non-dysplastic BE?

4) (Minor) The recovered cell populations from the normal stomach samples are a little concerning. The relative proportion (chief cells being the dominant cell population) of cells seems to be off from what should be expected based on standard gastric structure. Do the authors have a sense for why this may be? Could it be a problem in cell typing or cell processing (leading to loss of cells)? Similarly, from figure 3 it appears that the chief cell population strongly expresses MUC6. Is this known to be the case? I could be wrong, but I was under the impression it was more expressed in the mucus neck cells of the stomach (and SPEM).

5) (Minor) Given these were fresh tissue samples taken from a presumably recent endoscopies, I assume the long term progression status/outcome of the BE patients is unknown?

Reviewer 2:

Wenzel et al., Single cell analysis of Barrett's esophagus...

The objective of this study was to map GWAS risk alleles of Barrett's esophagus and of EAC to discrete cell populations in the distal esophagus and proximal stomach identified in silico via scRNAseq. Estimates of familial Barrett's and EAC seem to hover around 7-10% and reflux/BE concordance in monozygotic twins is reported to be 30-40%, suggesting genetic risks that have been supported by more recent GWAS studies. Tying these GWAS data to particular cell types is interesting and exciting for understanding the development of both BE and its progression to EAC. That being said, this reviewer, who has a fair grounding in the field, found it difficult to follow the logical paths to the key conclusions presented in this work despite multiple readings. While this might be attributed to a mild cognitive decline in this reviewer, the authors really need to clarify the nomenclature for the derived cell types, explain what they conclude from the trajectory studies, and most importantly inject some form of transparency in the association of the LD blocks of risk alleles and causal genes within to a particular cell type.

1. Starting with the GWAS links, there is the most minimal attempt to make sense of the associations in Ref. 9 (GWAS hits not presented in any figure, Ref. 9 is not available unless under subscription), no attempt to deconvolute the LD blocks for likely causal genes versus passenger genes. In lieu of this, general links between identified cell types and all associations presented in Fig5, 6 with MDD associations as a control. This is far more abstract than it needs to be and misses opportunities to identify causal genes within LD blocks that would be revealing for the disease.
2. The trajectory analysis wanders into an area of intense investigation by multiple laboratories and, from what I can gather, tie the origin of BE to gastric chief cells. How does this relate to the cells of origin cited within (Ref 7, 8) and other work? This trajectory is mentioned with much aplomb and yet there seems to be minimal discussion of the findings or their implications.
3. The nomenclature employed throughout for biopsies and cell types seem to be used interchangeably and at times (e.g. lines 136-143) is confusing and overwhelming for the rank and file. Similar baffling descriptions are seen in lines

129-135. There are many other examples where the reader, familiar with this high-profile field, is simply swamped.

I have no doubt that there are very interesting findings here that link risk alleles to specific cell types that will be relevant to understanding the emergence and progression of Barrett's esophagus, but more attention has to be vested to communicate this.

Authors' response to the first round of review

Point-by-point response to reviewers' comments:**Reviewer #1:**

Wenzel et al have performed scRNAseq on a series of BE and EAC patients with samples from the stomach, esophagus, and BE/EAC. Combining this data with previously identified GWAS risk loci, the authors suggest specific cell types that may be influencing BE formation or progression to EAC. The idea to link GWAS risk loci to nearby genes and determine which cells express those genes is an interesting idea and the authors findings point to both epithelial and stromal/inflammatory cells contributing to BE formation and progression. While maybe not completely novel, the idea of non-epithelial cells contributing to progression is definitely a newer idea and further refinement of which epithelial cells are major contributors is also an area of ongoing investigation. Overall, further refining cells that influence BE formation and progression is an important area. With the limited number of patients that can be analyzed through scRNAseq, this however seems more like a starting point than the definitive results the authors put forth in the conclusions. The inclusion of a second scRNAseq dataset definitely helps, however, it seems larger validation or functional studies to validate the findings would help solidify the findings. Without this, a portion of the discussion should be given to the limitations of the study.

We thank the reviewer for the overall positive assessment of our work. We agree that our findings are primarily based on statistical genetic approaches, relying on the integration of available GWAS with scRNA-seq data from biopsies of BE and EAC patients. In particular, we could prioritize cell types for BE and EAC associations based on the single-cell specific expression of genes showing an enrichment of the corresponding gene-signal at the level of genetic-loci associations. We acknowledge that functional validation will be necessary to confirm the potential involvement of specific cell types and genes in Barrett's esophagus (BE) and, in particular, its progression to esophageal adenocarcinoma (EAC). Unfortunately, currently no additional biopsies and tissue collections of BE patients with long term follow up data on progression status are available for further gene-expression analysis (and no comparable dataset is publicly available) but we hope that our study can provide a basis for future investigations and promote the generation of additional scRNA-seq datasets. We have also revised the discussion to better highlight that the current findings should be considered as hypotheses that require experimental validation. Below, we respond to the reviewer's comments point by point.

1) Could the findings be validated in a larger cohort of BE patients with known progression status utilizing a cheaper/high throughput technique like IHC/IF?

We agree with the reviewer that further validation of our findings in additional cohorts would be valuable. To strengthen the biological interpretation of our results, it would be important to perform gene expression analyses in a larger cohort of BE samples, ideally including both non-progressed BE and BE cases that progressed to EAC. Unfortunately, we do not currently have access to additional samples for follow-up analysis. However, we note that our approach by leveraging large-scale GWAS with robust genetics in combination with single-cell RNA-sequencing data across BE and EAC in esophageal tissues from independent samples already takes into account a signal from both genetic and expression levels. Specifically, the integration of these independent data sources allows us to prioritize cell types based on the enrichment of heritability explained by variants near highly specifically expressed genes using LD score regression. Nonetheless, we acknowledge that further studies with larger sample sizes, ideally including progression-stratified BE cohorts and complementary high-throughput expression assays such as IHC or spatial transcriptomics, will be important to validate and extend our findings. The collection of such samples requires substantial efforts since only 3.5% of BE progress to EAC (Jankowski et al. 2010 BMJ) and the exclusion of progression requires follow up periods of at least 10 years.

To reflect this limitation and provide this outlook, we added in line 347:

"Further functional experiments and studies with collections of BE tissue of progressing/non-progressing patients are necessary to elucidate this relationship and to investigate the prognostic value of individual cell type characteristics."

2) Overall the authors findings are interesting, however, I am not sure they fully support all of the conclusions. Without functional evidence, the authors need to be very careful on wording with interpretations. Some of their data is quite suggestive, but at least in my opinion, does not provide conclusive evidence of cause and effect. Many of the conclusions appear overstated and in my opinion, suffer from confounding associations with cause and effect. For example, statements such as "specific types of

fibroblasts and endothelial cells contribute to BE and EAC development, while dendritic cells and CD4+ memory T cells contribute exclusively to BE development (abstract)", seem exaggerated without providing any functional evidence or at least any refuting alternative explanations. There are many factors that can influence BE progression, it seems the methods used by the authors do a reasonable job at suggesting cell types influencing certain processes but it is not as clear that a negative result (no enrichment) means the cells are not contributing to BE progression or formation.

We agree that by combining scRNA-seq and GWAS data we are not able to prove the functional causality and can only suggest functional connections based on significant associations between transcriptomic and genetic signal. There are surely non-genetic factors that influence disease formation, especially in the earlier stages, e.g. BE being caused by chronic reflux and consequential inflammation. To strengthen this connection, functional studies of identified cell types are necessary. To better clarify the exploratory nature and scope of our analysis, we have now included statements in the Abstract, Results, and Discussion sections emphasizing that our findings are based on associations between GWAS signals and scRNA-seq expression data. We further acknowledge that functional validation will be necessary to support the biological hypotheses generated from these statistical associations:

- Abstract line 47 ff indicates now that the analyses suggest the conclusions.
- Results, section "Enrichment of genetic risk loci in cell type-specific gene expression profiles": the wording has been adjusted reflecting that our findings are indications for functional involvement rather than functionally validated. In addition: page 10, paragraph on fibroblasts/endothelial cells, line 242 added: "Further research is necessary to determine the role of this cell type in neoangiogenesis as a way of promoting tumor progression by non-malignant cells."
- Discussion: the general wording has been adjusted also here. In addition: page 14, line 347 ff: "Further functional experiments and studies with collections of BE tissue of progressing/non-progressing patients are necessary to elucidate this relationship and to investigate the prognostic value of individual cell type characteristics."

3) As above, The statement (line 281) "that germline genetic risk factors that control all steps of the cellular differentiation of the columnar BE epithelium contribute to EAC. This also includes the last step of columnar differentiation into differentiated cells that showed the strongest enrichment of EAC GWAS risk loci in the BE epithelium scRNA

Seq data set." needs reworded or further explained. 1) It is confusing as written and 2) from what I understand it to mean, it is too strong of a statement. Could it be possible that the effect of the loci/expressed gene occurs early, but just happens to have sustained or increased expression in differentiated cells? Again, without functional evidence these statements seem exaggerated.

We thank the reviewer for this comment. Indeed, the point in time when the expression of risk loci genes plays out its distinctive role is not precisely known, especially not by vertical data acquisition as in scRNA-seq. We can see through our trajectory analysis and the knowledge gathered by Nowicki-Osuch et al. that the intestinal metaplasia cell type (undifferentiated BE) has the closest transcriptomic relationship to cancer cells. An enrichment in risk loci genes in the expression profile of this cell type strongly suggests that the effect of these alteration can be active at any point of time when the given cell is in the cell type-typical or very similar state. We rephrased the respective section in the following way (now line 308 ff):

"Our data support the notion that differentiated and undifferentiated columnar cells are of relevance in EAC development. This is based on the transcriptomic enrichment of risk loci, where the expression profiles of differentiated columnar cells show the strongest enrichment."

4) While I am not an expert in this area, I would assume GWAS hits for MDD would effect genes expressed in nervous tissue/brain. Given the very different tissue type, it does not seem unreasonable none of the loci are near genes expressed in the stomach/esophagus. What happens if GWAS hits for a different cancer that is closer in tissue type but doesn't show overlap in GWAS loci is used (?maybe lung or breast adenocarcinomas) as a control?

We thank the reviewer for this suggestion. We agree that selecting a more biologically relevant cancer phenotype as a control might provide a more informative null model. Accordingly, we repeated our partitioned heritability enrichment analysis (LDSR) using GWAS summary statistics from a comparable powered GWAS of lung cancer in never-smokers, which includes 2,355 cases and 7,504 controls of European ancestry (GWAS Catalog study GCST004747). As shown in Table R1, we did not observe significant LDSR enrichment in any of the cell populations identified in our study.

Table R1. Partitioned heritability analysis using LD score regression (LDSR) analysis to assess the enrichment of risk variants from GWAS summary statistics for lung cancer in never-smokers (2,355 cases and 7,504 controls of European ancestry; GWAS Catalog study GCST004747).

Name	Enrichment value	p-	Coefficient	Coefficient_z.score
Chief.Cells	0.11		-2.20E-08	-1.583131346
Generic.myofibroblasts.CAFs	0.16		-2.21E-08	-1.400464594
Esophageal.fibroblasts.2	0.19		-1.91E-08	-1.318026961
Gastric.endothelial.cells	0.23		-1.63E-08	-1.215147147
CAFs	0.30		1.51E-08	1.028428569
Intestinal.metaplasia.cells	0.32		-1.56E-08	-1.00008376
Basophils	0.34		1.60E-08	0.963848552
CD8..NKT.like.cells	0.38		-1.38E-08	-0.871293014
Endothelial	0.40		1.28E-08	0.840433647
Esophagus.epithelial.cells	0.41		-1.29E-08	-0.822396182
EAC.04	0.43		1.43E-08	0.79764506
Misclassified	0.46		1.15E-08	0.744803124
Esophageal.endothelial.cells.1	0.50		-9.85E-09	-0.674849749
Esophageal.endothelial.cells.2	0.52		-8.93E-09	-0.648565873
Foveolar.cells.2	0.54		-9.26E-09	-0.614182003
Plasma.B.cells	0.56		-9.12E-09	-0.578948392
Lymphatic.endothelial.cells	0.60		-7.67E-09	-0.526319163
Parietal.cells	0.61		-7.67E-09	-0.509971215
Neuronal.cells	0.61		6.53E-09	0.506276423
Macrophages	0.62		-8.22E-09	-0.500098441
EAC.02	0.64		-7.19E-09	-0.473661221
Classical.Monocytes	0.75		-4.89E-09	-0.312983782
Esophageal.fibroblasts.1	0.79		3.53E-09	0.265935665
Enteroendocrine.Cells	0.79		3.97E-09	0.26045547
Pre.B.cells	0.84		-3.15E-09	-0.197721547
Gastric.fibroblasts.2	0.85		2.60E-09	0.188498256
BE.fibroblasts	0.86		-2.65E-09	-0.18260363
Plasmacytoid.dendritic.cells	0.86		3.03E-09	0.179952266
Foveolar.cells.1	0.86		-2.82E-09	-0.173935775
EAC.01	0.87		-2.82E-09	-0.168530121
Gastric.fibroblasts.1	0.89		-2.07E-09	-0.137817098
EAC.03	0.90		-2.02E-09	-0.122298855
Memory.CD4..T.cells	0.93		-1.36E-09	-0.090460055

We refer to this analysis in line 214 of the manuscript added the analysis as Supplementary Table 8.

We further examined the correlation between the rank of LDSR coefficient z-scores across cell populations using the BE/EAC GWAS summary statistics and those from the lung cancer GWAS. These z-scores quantify the strength and direction of the association between cell-type-specific annotations and disease heritability, with higher z-scores indicating stronger enrichment of GWAS signal in genes specific to a given cell population. This approach enables comparison of enrichment patterns independently of p-values, which can be influenced by factors such as the total heritability observed from the GWAS. As shown in Figure R1, we did not observe any significant correlation between the LDSR coefficient z-scores of cell populations derived from the lung cancer GWAS and those from the BE GWAS ($R = 0.11$, $p = 0.53$, Spearman's rank correlation) or the EAC GWAS ($R = -0.076$, $p = 0.67$, Spearman's rank correlation). Altogether, these results suggest that the cell-type-specific enrichment patterns observed in BE/EAC are specific and cannot be recapitulated in the lung cancer GWAS.

Figure R1. Distinct cell populations are enriched for risk variants in EAC and lung cancer. Comparison of LDSR coefficient z-scores across cell populations identified in our study using: A) BE GWAS summary statistics (x-axis) versus lung cancer GWAS summary statistics (y-axis), and B) EAC GWAS summary statistics (x-axis) versus lung cancer GWAS summary statistics (y-axis). The lung cancer GWAS summary statistics are from the GWAS Catalog (study GCST004747), based on genome-

wide association data for lung cancer in never-smokers of European ancestry (2,355 cases and 7,504 controls). The red dashed line represents the linear regression fit, and the gray shaded area indicates the 95% confidence interval of the fitted regression line, reflecting the uncertainty in the correlation between LDSR z-scores across traits. R represents spearman's rank correlation.

5) The GSEA (In 129-135) for EAC-01/03 vs EAC-02/04 had upregulation of gene sets for squamous esophageal cells (suprbasal cells). Could this be contamination of normal squamous esophageal cells in the cancer samples?

The concept of GEM-based single cells RNA-sequencing alters the concept of contamination within the formed clusters in comparison to classical RNA-seq. As cells are dissociated during the process of preparation for sequencing and then sequenced separately, the proposed contamination could only be based on a widespread doublet pairing of normal esophageal cells and cancer cells within single droplets on one hand, which is stochastically unlikely itself and also addressed in quality control. On the other hand, normal cells with transcriptomic features so similar to the cancer cells that they would cluster together, would still have a different CNV-profile which would change the data results in our inferCNV analysis.

In consequence, we must assume that the malignant cells tend to have a more squamous cell type as a biologically true finding.

6) This may be my lack of experience with LDSR, but how would a risk allele that decreases expression of a gene (for example a risk allele that decreases expression of a tumor suppressor pathway) factor into the analysis? Similarly, if a risk loci is linked to a gene that is widely expressed by multiple cell types will this not show up in the analysis?

We thank the reviewer for this comment. We would like to clarify that the applied LDSR partitioned heritability approach is based on mapping genetic signals to the genomic coordinates of genes that are highly and specifically expressed in single-cell data. This method does not incorporate eQTL directionality, meaning that genetic loci associated with either upregulation or downregulation of gene expression are treated equally in terms of genetic signal. Regarding the gene expression analysis, we focused on the top cell-type-specific genes for GWAS heritability enrichment using LDSR regression. Consequently,

only genes with high cell-type specificity are captured by this approach, while ubiquitously expressed genes—such as housekeeping genes or broadly expressed disease-related genes—are not expected to contribute to the observed enrichment. This strategy is deliberate, as our primary goal was to prioritize cell types characterized by highly specific expression patterns that show enrichment for GWAS heritability, rather than to perform a general mapping of GWAS loci to expression profiles.

7) Line 215: "pointing at blood vessels influencing disease formation in terms of neoangiogenesis" This statement may be a little strong. What evidence is there this is due to neoangiogenesis, especially if the cell type was predominately found in normal tissue and not in regions where the BE or EAC develop?

It is true that our data does not provide functional evidence that neoangiogenesis is the process through which endothelial cells confer risk for BA and EAC, respectively. However, our data does show that some endothelial cell types (esophageal endothelial cells-2 and gastric endothelial cells) show expression profiles in which an enrichment for GWAS risk loci are found for BE and EAC. Notably, the cluster of endothelial cells in which the large majority of tumor tissue sample-derived endothelial cells are included (esophageal endothelial cells-1) did not show an enrichment in GWAS risk loci, which suggests that esophageal endothelial cells-2 and gastric endothelial cells influence disease formation not at the time the tumor has already formed but at earlier stages. One plausible explanation would be neoangiogenesis, which we are unable to functionally proof using our approach. Therefore, we toned down the statement. We changed the respective section now on page 10 line 239 ff to

"In addition, endothelial cells of the esophageal endothelial cells-2 and gastric endothelial cells, the majority of which derived from normal esophagus and fundus, respectively, were significantly enriched for BE and EAC risk variants (Figure 5B) suggesting a role in disease formation. Further research is necessary to determine the role of this cell type in neoangiogenesis as a way of promoting tumor progression by non-malignant cells."

8) If the patients with and without cancer were separated, does the non-epithelial cells (fibroblasts, endothelial cells,...) still show enrichment for EAC GWAS risk variants? If the enrichment holds for the BE patients, given that the majority of patients with BE will

never progress to cancer, how does this affect the interpretation of the EAC GWAS hit findings?

We appreciate the reviewer for raising this very insightful point. The cell types of the single cell sequencing experiment have been defined using similarity-based clustering. Therefore, we expect for example esophageal endothelial cells-2 found in patients with BE and patients with EAC to be quite similar. Most of the non-cancer cells were identified in BE and EAC patients as shown for epithelial cells in Fig. 2B. However, some cell types are uniquely identified in either BE or EAC patients or are more abundant in one or the other patient category.

To investigate the point raised by the reviewer, we performed two complementary analyses. In the first analysis, we examined the transcriptomic profiles of cell populations shared between BE and EAC patients by calculating the Spearman's correlation of average gene expression levels across all genes between the two groups (if cell types were abundant in both sample origin groups). Here we observed that the expression profiles mostly showed very high correlations when comparing the expression profiles of the cell type cells from BE and EAC samples (Spearman's correlation coefficient ranging from 0.68 to 0.95, Figure R2). Among the epithelial cells, intestinal metaplasia cells from BE and EAC showed a very high correlation of $r=0.93$ (Figure R2B middle left), which we would expect based on the pivotal role in disease formation. This analysis supports the idea that the main characteristics of these cell types are patient-of-origin-independent.

A

Figure R2: Spearman correlation analysis of expression levels of individual cell types defined by BE or EAC patient-derived cells. Expression values of cells derived from BE patients are shown on the x-axis, expression values of cells derived from EAC patients are shown on the y-axis. Cell types with sufficient cells in both patient types are shown.

As a second analysis, we performed an additional LDSR enrichment analysis, where we compared the enrichment of EAC GWAS risk variants across different cell populations, separating BE and EAC patient samples as suggested by the reviewer. We focused specifically on the non-epithelial cell types that showed significant LDSR enrichment and here on the cell populations shared between BE and EAC patients. As shown in Table R2, esophageal endothelial cells-2 of both, BE and EAC patients, show enrichment although the FDR adjusted p-value is only suggestive of significance for BE patient-derived cells. Similarly, memory CD4+ T cells derived of BE and EAC patients show enrichment. This

finding suggests that EAC risk variants could, in principle, exert their influence on non-epithelial cells already within BE patients, potentially contributing to the progression toward cancer. Gastric endothelial cells do not reach FDR adjusted significance in this analysis.

We would also like to emphasize that in polygenic and multifactorial conditions such as BE and EAC, gene–environment interactions likely play a major role in modulating disease progression. Therefore, the presence of significant enrichments in BE patients, even though many of them do not progress to cancer, is a biologically plausible scenario.

Table R2. Enrichment p -values and their FDR-corrected values (Benjamini–Hochberg correction) for different cell populations, based on analyses performed on cells isolated from BE and EAC patients.

Cell population	Enrichment p -value		Enrichment p -value (FDR adjusted)	
	BE patients	EAC patients	BE patients	EAC patients
Memory CD4+ T cells	0.007963919	0.028128211	0.035497813	0.059772449
Esophageal endothelial cells-2	0.030481959	0.010147867	0.060963917	0.035497813
Gastric endothelial cells	0.499291721	0.037528784	0.499291721	0.070887704

We believe that this analysis shows that the differences between cells of a given cell type, but different sample origins, e.g. BE and EAC samples, are small enough for our conclusions to hold true.

9) In the opposite direction; Once an invasive cancer forms, even if small, it can exert an influence on the expression pattern and types of surrounding cells. This could affect expression profiles of both fibroblasts, endothelial cells, inflammatory cells, and even nearby non-neoplastic epithelial cells. If the EAC patients are contributing significantly to any of the findings, this seems like a potential caveat to the conclusions?

We thank the reviewer for pointing this out. This is related to point 8 above. There are indeed non-cancer cell types that exist in a cancer-induced state in particular or exclusively in EAC. A prime example is cancer associated fibroblasts (CAFs) where the highly plastic fibroblast acquired a state usually not found in non-cancer tissue. BE fibroblasts and esophageal fibroblasts-2 are characteristic for BE patient tissue. We extracted significantly more

basophils and pre-B cells from EAC tissue compared to BE tissue. Most cell types of our study, however, are found in both patient categories (Fig. 2B, Supplementary Fig. 2E, F, and Supplementary Table 2) and these cell types are comparable in both patient types (see our response to point 8).

To make this point clear to the reader, we added the following to the manuscript in line 99:

"While most non-cancer cell types were derived from both, BE and EAC patients, cancer associated fibroblasts (CAFs) were exclusively found in EAC patient tissue and BE fibroblasts as well as esophageal fibroblasts-2 exclusively in BE patient tissue. Basophils and Pre-B cells were more abundant in EAC patient tissue (Supplementary Table 2, Supplementary Note 1)."

To systematically assess whether the presence of cancer substantially alters the enrichment signal across cell populations, we compared the LDSR coefficient z-scores for different cell populations (see also reply to the point 4 on cancer type-specific effects and Figure R2 on cell type similarities for BE and EAC tissue). We calculated the absolute differences in coefficient z-scores for matching cell populations (e.g., intestinal metaplasia cells from BE vs. EAC patients) and compared the variability of these differences to those from non-matching pairs (e.g., fibroblast of BE patients vs. endothelial cells of EAC patients).

To statistically assess whether matching cell populations exhibit more consistent enrichment patterns (i.e., significantly lower variance), we performed both F-test and Levene's test to ensure robustness against non-normality in the data. Both tests showed that the variance of the difference in LDSR Coefficient z-scores was significantly lower for matching cell populations ($p = 0.02$ for the F-test and $p = 0.03$ for Levene's test), indicating that shared cell populations retain relatively stable GWAS enrichment profiles across BE and EAC conditions, despite the presence of cancer (Figure R3).

Figure R3. Boxplots showing the distribution of absolute differences in LDSR Coefficient z-scores between matching and non-matching cell population pairs from BE and EAC patients. Matching cell populations exhibit significantly lower variance in z-score differences, suggesting more consistent GWAS enrichment profiles across conditions.

Overall, we acknowledge that local tumor-induced changes in the microenvironment could still influence the transcriptomic or chromatin landscape of nearby cells, and we added the following sentences to Results (lines 229ff) to address these caveats:

"We also performed additional analyses to assess whether the presence of cancer substantially alters the enrichment signal across cell populations. By comparing LDSR-derived coefficient z-scores between matching and non-matching cell populations from BE and EAC patients, we found that matching cell populations exhibited significantly lower variance in enrichment signals ($p = 0.02$, F-test; $p = 0.03$, Levene's test, Supplementary Figure 8). These results suggest that, despite possible tumor-related remodeling, shared cell populations retain relatively stable genetic enrichment patterns. Nonetheless, tumor microenvironment may still contribute to shifts in gene regulation or cellular states, which is an important avenue for future studies."

10) From looking at the marker expression in Figure 3 and as stated by the authors, it appears the BE_GFN cluster would resemble non-intestinal (foveolar-like) metaplasia and the BE_EAC cluster would resemble intestinal metaplasia (true BE). Given the known differences in risk of progression associated with non-intestinal vs intestinal metaplasia and the enrichment of BE_EAC in patients with EAC in this cohort, could this be confounding some of the conclusions?

We reviewed the literature extensively to address this point. Indeed, foveolar cells-1 resemble foveolar cells with distinctive marker genes *MUC5AC*, *TFF1* and *TFF2*. As intestinal metaplasia cells show features of both goblet cell differentiation (e.g. *TFF3*, *MUC2*) and intestinal stem cells (e.g. *OLFM4*), the phenotype lies between differentiated and undifferentiated BE cells as found in Nowicki-Osuch et al 2021. The cells therefore can be described as true BE/intestinal metaplasia. Intestinal metaplasia and its influence on diagnostics and risk of progression is subject of controversy. In our data, we see that intestinal metaplasia cells are present in both, cells from BE and EAC samples, and given the proven risk of progression of patients diagnosed with intestinal metaplasia, we see this as evidence that intestinal metaplasia was the predecessor of EAC.

We added to the discussion on page 13, line 311:

"It is in line with the increased risk of BE (usually characterized by intestinal metaplasia) consisting of columnar cells to develop EAC ^{1,2}"

1) (Minor) Though to a lesser extent, BE can contain CNVs that are also seen in EAC. Were these clusters completely copy neutral? Can the authors define "large scale copy gains". By large scale do they mean covering a large genomic region or many copy gains (high copy gain) in a specific region?

We thank the reviewer to help us clarify the results of our analysis. By large scale copy gains, large genomic regions are meant. Figure 3 illustrates the genomic copy number differences between the cancer clusters and the clusters intestinal metaplasia cells and foveolar cells-1. While small regions of copy number alterations in intestinal metaplasia cells and foveolar cells-1 cannot be excluded with this analysis, no coherent copy number alterations can be observed in intestinal metaplasia cells and foveolar cells-1 in contrast to the cancer clusters. We rephrased on page 6 line 131 the findings on large copy gains:

15

"Notably, EAC-02 and EAC-04 shared three copy gains covering large regions on 6p, 8q, and 9q."

2) (Minor): For Sup Fig 6A, can the authors please explain what cells are in the BE/EAC transition zone besides the BE_EAC cluster? The results between BE/EAC and BE_EAC are slightly different, but it is hard to tell what other cells are in that circled area besides BE_EAC.

We agree with the reviewer that the inclusion of small cell populations in the BE/EAC transition zone for differential expression analysis might not be focused enough. We sharpened the analysis in the way that we focus on the dominating cell type of the transition zone now, with the new nomenclature intestinal metaplasia cells (former BE-EAC). Intestinal metaplasia cells are compared with the broad group of epithelial cells of BE and gastric fundus (Supp. Fig. 6A), foveolar cells-1 and foveolar cells-2 are compared (Supp. Fig. 6B), and intestinal metaplasia cells are compared with the cell type foveolar cells-1 (Supp. Fig. 6C). We changed Supp. Fig. 6 and Supp. Table 6 accordingly.

3) (Minor) Were the BE patients with exclusively non-dysplastic BE?

Only BE patient 4 had low-grade dysplasia included in the pathology report, all others did not have dysplasia. We added this information to Supplementary Table 1.

4) (Minor) The recovered cell populations from the normal stomach samples are a little concerning. The relative proportion (chief cells being the dominant cell population) of cells seems to be off from what should be expected based on standard gastric structure. Do the authors have a sense for why this may be? Could it be a problem in cell typing or cell processing (leading to loss of cells)? Similarly, from figure 3 it appears that the chief cell population strongly expresses MUC6. Is this known to be the case? I could be wrong, but I was under the impression it was more expressed in the mucus neck cells of the stomach (and SPEM).

We see an expression of *MUC6* in chief cells (Fig. R4). Chief cells were determined by the expression of chief cell-specific genes such as progastrin-species. Indeed, it was reported previously that *MUC6* is mainly expressed in mucous neck cells. As in intestinal metaplasia

cells, we see a transcriptomic overlap in this regard. It is technically extremely unlikely that this is due to contamination (i.e. cell doublets instead of single cells), it appears that chief cells do show large quantities of *MUC6*-mRNA. Obviously, we cannot measure the protein quantities. Busslinger et al. 2021 (Cell Rep) found *MUC6* expressed in chief cells albeit at much smaller quantities compared to neck cells. The lack of a mucous neck cell population seems to be due to a bias in the recovery of these cells along the process between tissue dissociation and cell type recovery. Therefore, we do not derive conclusions regarding the relative quantities of cells within a tissue sample.

Figure R4: UMAP projection of epithelial cells indicating expression levels of *MUC6*. Several cell types show *MUC6* expression to various extents.

5) (Minor) Given these were fresh tissue samples taken from a presumably recent endoscopies, I assume the long term progression status/outcome of the BE patients is unknown?

Yes, this is correct. Since we used a protocol that requires fresh tissue for dissociation and single cell preparation, the patients have been recently in the clinic and we have only a limited time of follow up that is inconclusive with regards to progression status.

Reviewer #2: Wenzel et al., Single cell analysis of Barrett's esophagus.

The objective of this study was to map GWAS risk alleles of Barrett's esophagus and of EAC to discrete cell populations in the distal esophagus and proximal stomach identified in silico via scRNAseq. Estimates of familial Barrett's and EAC seem to hover around 7-10% and reflux/BE concordance in monozygotic twins is reported to be 30-40%, suggesting genetic risks that have been supported by more recent GWAS studies. Tying these GWAS data to particular cell types is interesting and exciting for understanding the development of both BE and its progression to EAC. That being said, this reviewer, who has a fair grounding in the field, found it difficult to follow the logical paths to the key conclusions presented in this work despite multiple readings. While this might be attributed to a mild cognitive decline in this reviewer, the authors really need to clarify the nomenclature for the derived cell types, explain what they conclude from the trajectory studies, and most importantly inject some form of transparency in the association of the LD blocks of risk alleles and causal genes within to a particular cell type.

1. Starting with the GWAS links, there is the most minimal attempt to make sense of the associations in Ref. 9 (GWAS hits not presented in any figure, Ref. 9 is not available unless under subscription), no attempt to deconvolute the LD blocks for likely causal genes versus passenger genes. In lieu of this, general links between identified cell types and all associations presented in Fig5, 6 with MDD associations as a control. This is far more abstract than it needs to be and misses opportunities to identify causal genes within LD blocks that would be revealing for the disease.

We thank the reviewer for the comment. In our analysis, we primarily focused on cell-type prioritization by integrating GWAS signals with single-cell RNA-seq data, specifically modeling polygenic enrichment across the most highly and specifically expressed genes in each cell type. Our approach was therefore intended to capture broader polygenic effects rather than to directly assign functional relevance to individual GWAS loci. We acknowledge that GWAS loci often encompass multiple genes due to linkage disequilibrium (LD), and that these genes may exhibit correlated expression patterns. As a result, multiple genes within a locus may appear associated, even though only one—or a few—may represent the true causal effector(s). The remaining associations may be spurious, driven by LD structure or coincidental transcriptional co-regulation. This complexity introduces pleiotropy and poses challenges for identifying causal genes.

While our framework was not specifically designed to resolve causal genes at the locus level, it can be extended to support gene-level prioritization by leveraging the specificity of expression across relevant cell types. To support this, we have now added Supplementary Tables 9 and 10 listing all genome-wide significant loci for both BE and EAC, the genes located within those loci, and the prioritized genes based on highly cell-type-specific expression in the associated cell types identified through polygenic enrichment analysis. We hope these annotations will help guide future integrative analyses focused on fine-mapping and functional validation of the identified genes.

We added in line 216: "By focusing on single genes, we were able to prioritize candidate genes located within GWAS loci for EAC and BE (Supplementary Tables 9 and 10, respectively)."

Ref 9 (Schröder et al. 2023 Gut) is provided together with the submission.

2. The trajectory analysis wanders into an area of intense investigation by multiple laboratories and, from what I can gather, tie the origin of BE to gastric chief cells. How does this relate to the cells of origin cited within (Ref 7, 8) and other work? This trajectory is mentioned with much aplomb and yet there seems to be minimal discussion of the findings or their implications.

A trajectory analysis in general does not prove a trajectory regarding development/time, it calculates gradual steps in the transcriptomic space, inferring a trajectory that may be interpreted as developmental route, otherwise it is more conceivable as a path along similarities of transcriptomes. In our data, we see foveolar cells-1 as most similar to the intestinal metaplasia cells. With the limitations mentioned above, our data suggests that foveolar cells-1 are the cell of origin of BE. This is similar to the finding of Nowicki-Osuch and colleagues (Nowicki-Osuch et al. 2021 Science) using single cell RNA-sequencing and epigenetic analyses where various subtypes of BE cells show "strongest similarities to gastric cardia cells", including columnar undifferentiated, -intermediate, and -differentiated, as well as foveolar intermediate, and -differentiated cells. Owen and colleagues (Owen et al. 2018 Nat Commun) suggest cells of the submucosal gland as the cell of origin based on their scRNA-seq data with *LEFTY1* as a marker for both, BE and esophageal submucosal glands. While our results are in better agreement with the study of Nowicki-Osuch et al. it is difficult to exclude the submucosal glands with its specific cell types as we did not identify the respective cells in our data and we do not know where they would be placed in the trajectory analysis. Among the epithelial non-cancer cells, intestinal metaplasia cells show

20

the strongest expression of *LEFTY1* (Figure R5). Moving along the trajectory in the transcriptomic space away from cancer cells we find chief cells as the transcriptomic 'neighbors' of foveolar cells-1. Further down in our trajectory, the intestinal metaplasia cell type is closest to cancer cells. The intestinal metaplasia cells do not show evidence for large copy number variations and are therefore likely non-malignant, providing the putative 'cell of origin' of EAC, as was discussed by Nowicki-Osuch et al.

Figure R5: Expression of *LEFTY1* in epithelial cells. Top: UMAP of epithelial cells with color coding for *LEFTY1* expression. Bottom: Violin plots indicating *LEFTY1* expression (y-axis) for indicated cell types (x-axis). *LEFTY1* expression is very specific for IM-BE (intestinal metaplasia cells) in non-malignant cells as ascertained by CNV profiles. *LEFTY1* is also highly expressed in cells from EAC-01. FC1 and FC2, foveolar cells-1 and -2.

We address the relatedness of the relevant cell types more explicitly in the results section "Characterization of epithelial cells and detection of typical genomic aberrations in EAC" on page 5 lines 114ff.

"We identified a group of cells that show signs of goblet cell differentiation which defines intestinal metaplasia¹, but also expression of the gastrointestinal stem cell marker *OLFM4*^{2,3}, setting this cell type close to the undifferentiated BE cell type implicated by Nowicki-Osuch et al.⁴ to be the cell of origin of EAC."

Further, we expanded in the results under "Trajectory analysis implies a route to BE and EAC" on page 7, lines 165ff:

"To investigate the developmental relationship among epithelial cell types, we performed a trajectory analysis using *monocle3* based on transcriptomic similarity. Our analysis placed gastric chief cells close to foveolar cells-1 originating from gastric fundus and BE tissue, leading to the group of intestinal metaplasia cells, which is consisting of cells with goblet cell differentiation markers, but also expressing the intestinal stem cell marker *OLFM4*. The trajectory closely connected foveolar cells-1 with intestinal metaplasia cells, suggesting foveolar cells-1 as the cell of origin for BE (Figure 4B). This is in agreement with the columnar non-squamous phenotype of BE cells and supports the model of gastric cardia cells, located next to the fundus, as the origin of BE⁸. Further, the trajectory connected intestinal metaplasia cells to the malignant cells resembling the dysplastic developmental route of EAC (Figure 4B)."

3. The nomenclature employed throughout for biopsies and cell types seem to be used interchangeably and at times (e.g. lines 136-143) is confusing and overwhelming for the rank and file. Similar baffling descriptions are seen in lines 129-135. There are many other examples where the reader, familiar with this high-profile field, is simply swamped.

I have no doubt that there are very interesting findings here that link risk alleles to specific cell types that will be relevant to understanding the emergence and progression of Barrett's esophagus, but more attention has to be vested to communicate this.

We thank the reviewer for pointing out the complicated annotations of the identified cell types. After revision of the text and the literature, we were able to classify cell types that we

originally named BE-GFN and GFN as two subtypes of foveolar cells and BE-EAC as a mixture of differentiated/goblet cell-like and undifferentiated BE cells, following the reasoning of Nowicki-Osuch et al. (2021), which we describe now as intestinal metaplasia cells. We spell out cell types throughout the manuscript instead of using abbreviations. We are positive that the changed nomenclature reduces the risk for misunderstandings significantly.

The new nomenclature is as follows:

Old nomenclature	New nomenclature
BE Fibroblasts	BE fibroblasts
BE GFN	Foveolar cells-1
GFN	Foveolar cells-2
BE EAC	Intestinal metaplasia cells
EAC 01	EAC-01
EAC 02	EAC-02
EAC 03	EAC-03
EAC 04	EAC-04
EN Fibroblasts 1	Esophageal fibroblasts-1
EN Fibroblasts 2	Esophageal fibroblasts-2
ESO EC 1	Esophageal endothelial cells-1
ESO EC 2	Esophageal endothelial cells-2
Esophagus Normal	Esophagus epithelial cells
GFN EC	Gastric endothelial cells
GFN Fibroblasts 1	Gastric fibroblasts-1
GFN Fibroblasts 2	Gastric fibroblasts-2

References

Frankell AM, Jammula S, Li X, Contino G, Killcoyne S, Abbas S, Perner J, Bower L, Devonshire G, Ococks E et al. 2019. The landscape of selection in 551 esophageal adenocarcinomas defines genomic biomarkers for the clinic. *Nat Genet* 51: 506-516.

Schroder J, Chegwiddden L, Maj C, Gehlen J, Speller J, Bohmer AC, Borisov O, Hess T, Kreuser N, Venerito M et al. 2023. GWAS meta-analysis of 16 790 patients with Barrett's oesophagus and oesophageal adenocarcinoma identifies 16 novel genetic risk loci and provides insights into disease aetiology beyond the single marker level. *Gut* 72: 612-623.

Referees' report, second round of review

Reviewer 1:

The authors have responded well to all of my original questions. I think overall the manuscript is improved and the conclusions better match the data. Below are two minor suggestions that may be worth considering.

1) "the cluster of endothelial cells in which the large majority of tumor tissue sample-derived endothelial cells are included (esophageal endothelial cells-1) did not show an enrichment in GWAS risk loci, which suggests that esophageal endothelial cells-2 and gastric endothelial cells influence disease formation not at the time the tumor has already formed but at earlier stages." I completely agree with this statement. Are genes with known association with neoangiogenesis upregulated in esophageal endothelial cells-2 and gastric endothelial cells compared to esophageal endothelial cells-1? If so, that would give some suggestion that neoangiogenesis is playing a part.

2) "Intestinal metaplasia cells lacked SCNAs (see above) and had a higher metabolic turnover (oxidative phosphorylation), thereby connecting a hallmark of cancer with a non-cancer cell type." Suggest adding "detected" before SCNAs.

Authors' response to the second round of review

Response to reviewer comments

Reviewer #1: The authors have responded well to all of my original questions. I think overall the manuscript is improved and the conclusions better match the data. Below are two minor suggestions that may be worth considering.

1) "the cluster of endothelial cells in which the large majority of tumor tissue sample-derived endothelial cells are included (esophageal endothelial cells-1) did not show an enrichment in GWAS risk loci, which suggests that esophageal endothelial cells-2 and gastric endothelial cells influence disease formation not at the time the tumor has already formed but at earlier stages." I completely agree with this statement. Are genes with known association with neoangiogenesis upregulated in esophageal endothelial cells-2 and gastric endothelial cells compared to esophageal endothelial cells-1? If so, that would give some suggestion that neoangiogenesis is playing a part.

Response:

We performed a differential expression analysis between esophageal endothelial cells-2 and gastric endothelial cells compared to esophageal endothelial cells-1 and no markers for neoangiogenesis were found to be significantly differentially expressed presumably due to functional similarities between different endothelial cell types. In addition, we performed gene set enrichment analysis and found no terms of neoangiogenesis to be enriched. Again, since we compare quite similar cell types, it is well possible that there is no statistically significant pathway enrichment although one endothelial cell type is contributing while another is not contributing to disease development. The cited sentence of the reviewer was derived from our point-by-point-response. We did not change the respective sentence in the manuscript as this remains valid "Further research is necessary to determine the role of this cell type in neoangiogenesis as a way of promoting tumor progression by non-malignant cells."

2) "Intestinal metaplasia cells lacked SCNAs (see above) and had a higher metabolic turnover (oxidative phosphorylation), thereby connecting a hallmark of cancer with a non-cancer cell type." Suggest adding "detected" before SCNAs.

Response:

We added "detected" as suggested by the reviewer.